

# Properties and evolution of a submesoscale cyclonic spiral

Reiner Onken[1] and Burkard Baschek[2]

[1]Helmholtz-Zentrum hereon, Max-Planck-Straße 1, 21502 Geesthacht, Germany
[2]Deutsches Meeresmuseum, Katharinenberg 14–20, 18349 Stralsund, Germany

*Correspondence to:* Reiner.Onken@hereon.de

**Abstract.** The evolution of a submesoscale cyclonic spiral of 1 km in diameter is simulated with ROMS (Regional Ocean Modeling System) using 33.3 m horizontal resolution in a triple-nested configuration. The generation of the spiral starts from a dense filament that is rolled into a vortex and detaches from the filament. During spin-up, extreme values are attained by various quantities, that are organized in single-arm and multi-arm spirals. The spin-down starts when the cyclone separates from the filament. At the same time, the horizontal speed develops a dipole-like pattern and isotachs form closed contours around the vortex center. The amplitudes of most quantities decrease significantly, but the instantaneous vertical velocity $w$ exhibits high-frequency oscillations and more pronounced extremes than during spin-up. The oscillations are due to vortex Rossby waves (VRWs), that circle the eddy counterclockwise and generate multi-arm spirals with alternating signs by means of azimuthal vorticity advection. Experiments with virtual surface drifters and isopycnal floats indicate downwelling everywhere near the surface. The downwelling is most intense in the center of the spiral at all depth levels, leading to a radial outflow in the thermocline and weak upwelling at the periphery. This overturning circulation is driven by convergent near-surface flow and associated subduction of isopycnals. While the downwelling in the center may support the export of particulate organic carbon from the mixed layer into the main thermocline, the upwelling at the periphery effectuates an upward isopycnal transport of nutrients, enhancing the growth of phytoplankton in the euphotic zone.

## 1 Introduction

This paper describes the physical properties of a submesoscale cyclonic eddy and its evolution. In the context of this article, the authors prefer the expression "spiral", because the feature under investigation develops from a filament that rolls up in a spiraliform vortex, in contrast to a "ring" that becomes strangulated from an unstable front.

Spirals have been described comprehensively by Munk et al. (2000) and Munk (2001) as "10–25 km in size and overwhelmingly cyclonic" and "linear features ... wound into spirals in vortices associated with horizontal shear instability". In contrast, Buckingham et al. (2017) observed cyclonic vortices at the edge of an ocean front resulting from "mesoscale stirring, filamentation, and subsequent frontal instability", and concluded that mixed-layer or submesoscale baroclinic instability is a more plausible explanation for their generation, as already previously found by Eldevik and Dysthe (2002). Meanwhile, due to improved measurement techniques, smaller spiraliform patterns were observed. The width of the spiral arms is tens to hundreds of meters, while the size of the vortex is $\mathcal{O}(1\ \text{km})$ (Marmorino et al., 2018; McWilliams, 2019). Hence, cyclonic spirals with evolutional time scales of hours to days are typical elements of the submesoscale waveband.



Due to the small spatiotemporal scales, it is rather difficult to observe submesoscale spirals in situ. Therefore, the actual knowledge of the physical properties of the spirals is largely shaped by their signature at the sea surface detected by remote sensing techniques. The corresponding observations are mostly limited to images of proxy parameters such as sun-glint (Scully-Power, 1986), surface roughness (Karimova and Gade, 2016), phytoplankton (Fig. 1), or surfactants like oil slicks (D'Asaro

et al., 2018). Namely, such snapshots allow the identification of sharp convergence lines (Fig. 1) suggesting downwelling in submesoscale fronts and spirals according to theoretical studies, but they cannot describe the formation and the decay of the spirals and the associated three-dimensional circulation patterns. Some more insight is provided by in-situ observations of D'Asaro et al. (2018) of a cyclonic vortex about 10 km in diameter, concerning the internal mass field, kinematic properties and the clustering and dispersion of surface drifters. The SubEx campaign of Marmorino et al. (2018) focused on a much

smaller cyclonic eddy, the diameter of which was around 1 km. From airborne infrared measurements and nearly simultaneous observations with ADCP (Acoustic Doppler Current Profiler) and a Towed Instrument Array, kinematic quantities such as horizontal velocity, vorticity, horizontal strain, and divergence were estimated. In the framework of the same experiment, Ohlmann et al. (2017) computed divergence and vorticity distributions from surface drifters. To our knowledge, only the latter 3 citations provide concrete findings about the properties of submesoscale cyclonic spirals, but the perceptions are still

rather fragmentary. Before describing the evolution of a submesoscale spiral, we will therefore compile in the following some information concerning the corresponding patterns in mesoscale, primarily cyclonic eddies. Namely, the spatiotemporal scales of such eddies are several magnitudes larger, but sometimes they exhibit striking similarities to submesoscale spirals.

A comprehensive observational study during the formation of a Southern Ocean cyclonic mesoscale eddy was conducted by Adams et al. (2017), using drifters, Seasoar (a towed, undulating underwater vehicle), and ADCP. While the drifters orbited

the cyclone along its main front, cross-sections perpendicular to the drifters' paths were carried out simultaneously with the other instruments. The collocated Lagrangian drifter velocities and the Eulerian ADCP measurements showed an asymmetric ringlike circulation with high horizontal velocities in the north and weaker velocities in the south. Furthermore, regions of diffluent and confluent cross-frontal velocities were identified. Upwelling was associated with diffluent and downwelling with confluent flow patterns. In another study, Buongiorno Nardelli (2013) described the evolution of a cyclonic mesoscale eddy in

the Agulhas Return Current. The vertical circulation pattern was dominated by azimuthal oscillations, known as vortex Rossby waves (McWilliams et al., 2003). In a purely numerical study, Gula et al. (2016) described in detail the structure of various quantities inside a cyclonic Gulf Stream eddy. Further information may be gained from investigations of anticyclonic mesoscale eddies. For instance, dipolar and multipolar appearances of the vertical velocity were observed by Barceló-Llull et al. (2017), and are described in the numerical study of Estrada-Allis et al. (2019). The complex structure of the vertical velocity field was

subject of the investigations of Koszalka et al. (2009), Brannigan (2016) and Brannigan et al. (2017). Amongst others, they revealed spiraliform patterns and closely spaced cells of either sign at the periphery of an anticyclone. Similar to Gula et al. (2016) in the case of a cyclonic eddy (see above), Zhong et al. (2017) investigated various quantities in an anticyclonic eddy, with special emphasis of the vertical circulation pattern.

In order to deepen our knowledge of the evolution and physical properties of submesoscale vortices, the Regional Ocean

Modeling System (ROMS) is applied in a triple offline-nested configuration for a subregion of the Baltic Sea, located to the



**Figure 1.** Cyanobacteria bloom in the Baltic Sea (https://www.esa.int/ESA_Multimedia/Images/2015/09/Algae_bloom, last access 16 September 2020). The georeferencing was performed by the authors.





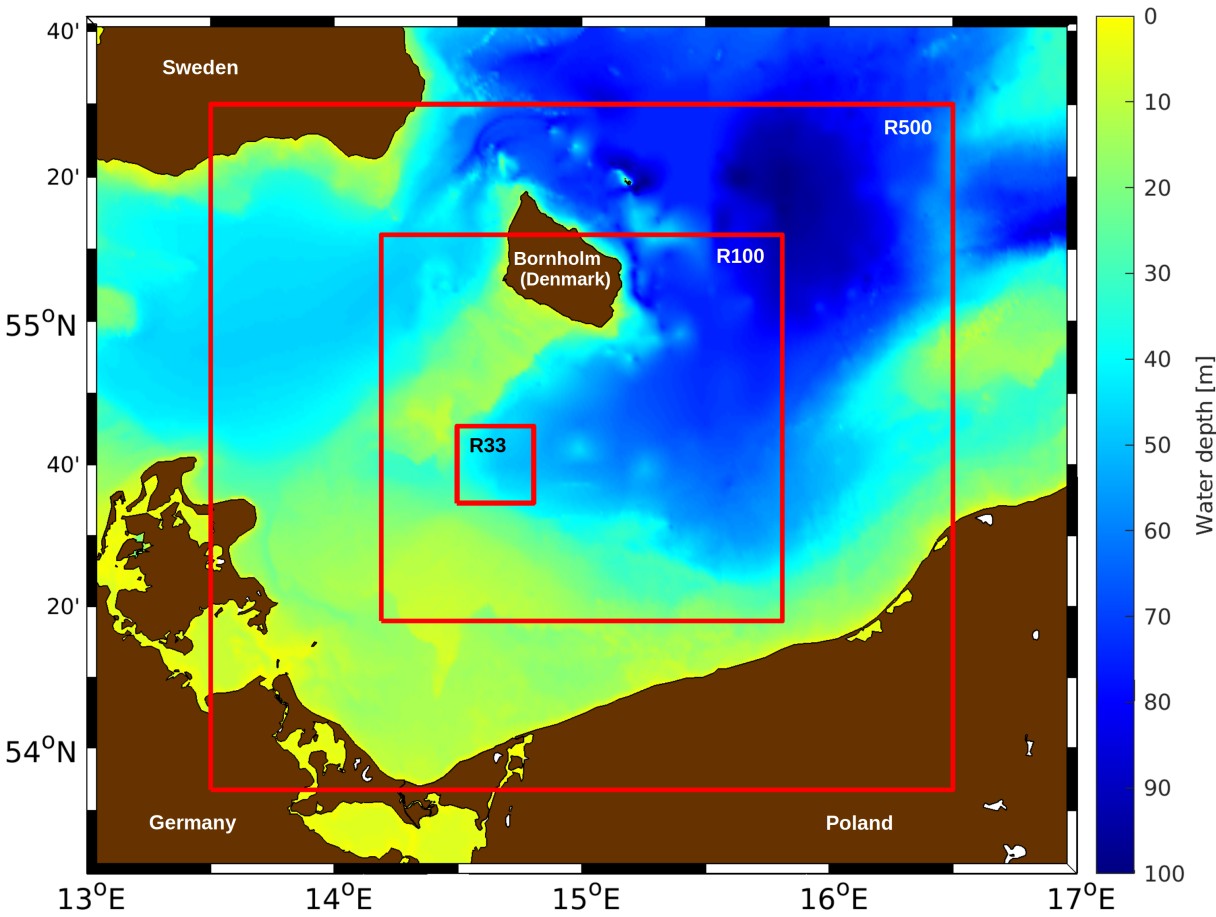

**Figure 2.** Arrangement of the nested ROMS domains in the western Baltic Sea.

south of the island of Bornholm (Fig. 2). The first nest with 500-m horizontal resolution (R500) is nested into the operational HIROMB-BOOS model (Berg, 2012, for details see Onken et al. (2020a)) and creates the mesoscale background for June 2016. Submesoscale structures develop rapidly in the second nest with 100-m resolution (R100), amongst others several cyclonic spirals with diameters around 1 km (Onken et al., 2020a). One of these spirals will be investigated in detail in this article by

5 means of a third nest with 33.3-m resolution, referred to as R33.

The R33 setup and the formation of the spiral are described in the following section. Kinematical and dynamical properties of the spiral are shown in Section 3. The impact of those properties on the behavior of Lagrangian drifters and floats is demonstrated in Section 4, followed by the Discussion and Conclusions. All time specifications refer to the year 2016 (unless stated otherwise) and are given in UTC (Universal Time Coordinated).





## 2 Model setup and formation of a spiral

ROMS is a hydrostatic, free-surface, primitive equations model. Its algorithms are described in detail in Shchepetkin and McWilliams (2005). Both in R100 and R33, the primitive equations in the vertical are discretized in 10 layers over a variable topography using stretched terrain-following coordinates, so-called s-coordinates (Song and Haidvogel, 1994). In the horizon-

tal, spherical coordinates are used. Biharmonic mixing along isopycnic surfaces is applied to the tracers using the diffusivity coefficients 1000 $m^4$ $s^{-1}$ in R100 and 50 $m^4$ $s^{-1}$ in R33. For R100, mono-harmonic mixing of momentum with an eddy viscosity coefficient $A_H^M = 10^{-2}$ $m^2$ $s^{-1}$ is used, while no explicit mixing is specified in R33. The vertical mixing of momentum and tracers is parameterized with the interior closure by Large et al. (1994). For bottom friction, a quadratic law is applied, and the pressure gradient term is computed using the standard density Jacobian algorithm by Shchepetkin and Williams (2001). No

atmospheric forcing is applied to the sea surface in R100 and R33, i.e. the fluxes of momentum, heat, and fresh water are set to zero. The reasons for this setup are described in detail in Onken et al. (2020a) and in Section 5.1 below.

Our special interest applies to a cyclonic spiral that starts to form in R100 from a northward intrusion of a filament on 24 June (see Fig. 12 in Onken et al. (2020a) and the animation in Onken et al. (2020b)). On the following day, the reeling begins, driven by a pre-existing barotropic circulation. At the same time, strong convergence of the horizontal currents occurs in the

"umbilical" of the spiral. On 26 June, the spiral begins to separate from the filament and the convergence weakens. One day later, the spiral is completely detached. In order to investigate this process in more detail, R100 was re-initialized from R500 and integrated until 28 June, providing temporally high-resolution boundary conditions in 3-hour intervals for R33. On 24 June, R33 was initialized from R100 and integrated from then until 28 June as well. For R33, a baroclinic time step of 20 s was used, while the prognostic variables along the open boundaries were updated by R100 at each time step by linear interpolation

in time. In order to demonstrate that the physics in the interior of the R33 domain was not contaminated by the boundary conditions, Fig. 3 shows the potential density anomaly $\rho$ (referred to as "density" in the following) and the horizontal currents in the top layer on 24–28 June for R100 and R33. On 24 June (panels a, f), the fields of both nests look identical, hence the downscaling appears to be perfect. First noticeable differences between the nests appear on 25 June in the northwest sector. The disagreements increase on 26 and 27 June, and additional disparities to the south of the spiral appear, but the spiral is

not yet affected. Finally on 28 June, the R33 patterns deviate significantly from R100, and also the spiral exhibits differences; in R100, its shape is more elliptical and the high-density core is better preserved in R33. Overall, the nesting procedure does apparently a good job.

## 3 Properties of the spiral

In this section, various properties of the spiral are shown by means of magnified images of the R33 domain. The images are

centered at the density maximum of the spiral, their meridional width is 2 arc minutes ($\widehat{=}$2 nautical miles $\approx$ 3704 m), and the zonal width is 4 arc minutes ($\approx$ 4354 m).





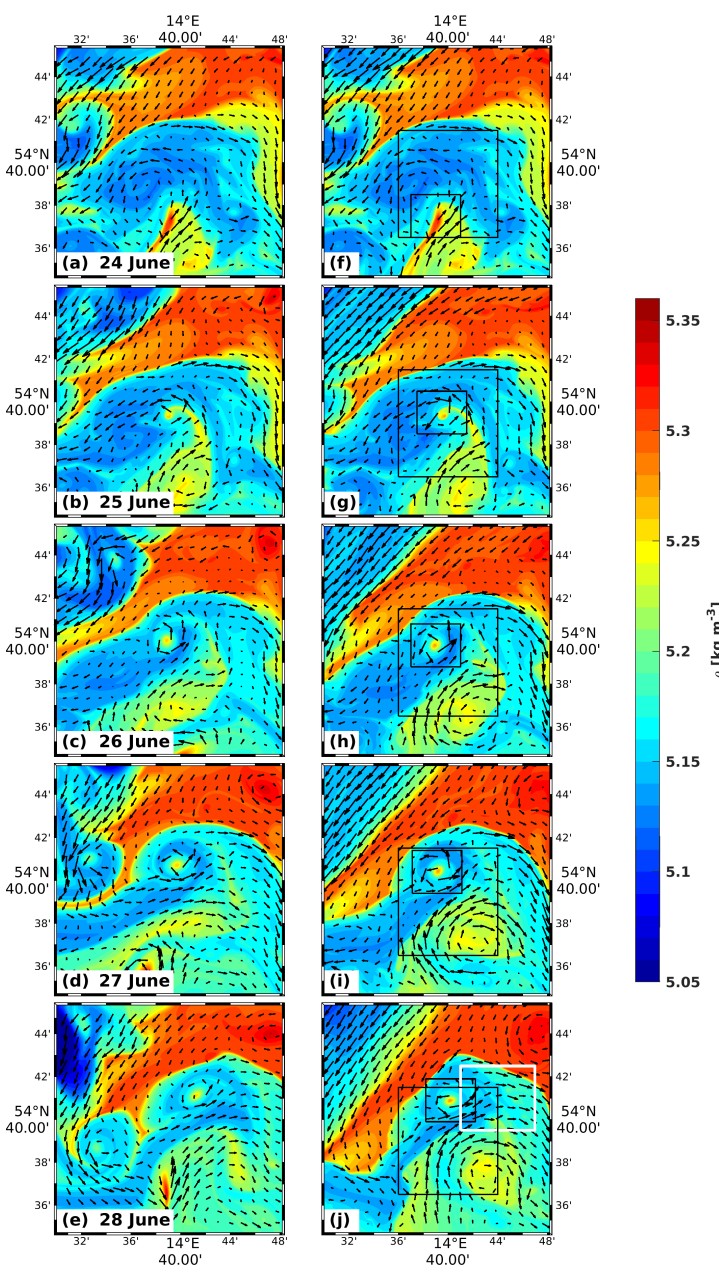

**Figure 3.** Top-layer density $\rho$ and horizontal velocity $V$ on 24 – 28 June in (a) – (e) R100 and (f) – (j) R33. Vectors are drawn at 1-km resolution. For R100, only the area covered by the R33 domain is shown. The small black boxes in the right column indicate the zoomed areas shown in Figs. 4, 5, 6, 7, 8. The big black boxes refer to the zoomed areas in Figs. 12 and 15, and the white box in panel j marks the zooms displayed in Fig. 10.





### 3.1 Fronts, frontogenesis and frontolysis

In the framework of this article, the term "front" denotes density fronts, i.e. regions with enhanced gradients of the horizontal density $\rho$. A measure of the strength of a front is $|\boldsymbol{\nabla}\rho|$. The frontal tendency $F = d|\boldsymbol{\nabla}\rho|/dt$ indicates whether a front intensifies (frontogenesis, $F > 0$) or weakens (frontolysis, $F < 0$). The enlarged view of the evolution of the surface density in Fig. 4a –

e reveals details that are not visible in Fig. 3f – j, e.g. the asymmetry of the filament (panel a) and the egg-shaped high-density core of the spiral (panels b – e). The development of $|\boldsymbol{\nabla}\rho|$ (panels f – j) indicates that on 24 June, the strongest gradients of about $6 \times 10^{-4}$ kg m$^{-4}$ are located at the western flank and close to the "head" of the filament in the north, while the gradients on the eastern flank are weaker. One day later, the maximum gradient has almost doubled and attains an all-time high of more than $11 \times 10^{-4}$ kg m$^{-4}$ in the hook-shaped structure evolving later on into a spiraliform pattern. The strong enhancement of

the density gradients during the first two days of the integration and the subsequent slackening is also reflected by the frontal tendency (panels k – o) exhibiting intense frontogenesis of $59 \times 10^{-12}$ kg$^{-2}$ m$^{-8}$ s$^{-1}$ and $64 \times 10^{-12}$ kg$^{-2}$ m$^{-8}$ s$^{-1}$ in the red patches of panels l and m, respectively. However, at the same time, extreme weakening of up to $\approx -48 \times 10^{-12}$ kg$^{-2}$ m$^{-8}$ s$^{-1}$ of the gradients takes place in the blue patches. Further analyses of the components of $F$ (see Hoskins (1982); Capet et al. (2008); Onken et al. (2020a)) have shown that at this stage the major contribution to the frontogenetic tendency comes from

$Q_h$, the straining deformation of the horizontal velocity, while the vertical straining is the main driver of frontolytic processes. The $Q_h$ pattern of 25 June (panel q) resembles closely that of Gula (2016, their Fig. 10f); this is the strong frontogenetic contribution on the upstream face of the spiral and the frontolytic sector at the tip of the hook.

### 3.2 Horizontal velocity

The total horizontal velocity $\boldsymbol{V}$ is the sum of the geostrophic and the ageostrophic velocity,

$$\boldsymbol{V} = \boldsymbol{V}_{geo} + \boldsymbol{V}_{ageo}. \tag{1}$$

On 24 June (Fig. 5a), the northward progression of the filament is perceptible by $|\boldsymbol{V}| > 5$ cm s$^{-1}$, where the maximum speed is close to 13 cm s$^{-1}$ at the eastern flank of the high-density core (cf. Fig. 4a). A closed cyclonic circulation pattern with two speed maxima develops on 25 June (panel b). Both maxima are discernible until 27 June when the spiral is completely detached and the highest speed in the southeast maximum drops to $< 11$ cm s$^{-1}$. Finally on 28 June (panel e), the latter maximum recovered

with maximum speeds close to 13 cm s$^{-1}$, while the other maximum in the northwest almost diminished. A similar dipole-like circulation pattern was observed by Adams et al. (2017) in a mesoscale eddy. Multiple maxima of the azimuthal speed are also reported by Buongiorno Nardelli (2013).

The geostrophic velocity (Fig. 5f – j) largely resembles the pattern of the total velocity $\boldsymbol{V}$. Differences of the directions are not distinguishable with the naked eye in contrast to minor visible discrepancies of the speed that are caused by the ageostrophic

velocity. On average, $|\boldsymbol{V}_{ageo}|$ (panels k – o) arrives at about 10% of $|\boldsymbol{V}_{geo}|$. Extreme values of 2.7 cm s$^{-1}$ and 2.6 cm s$^{-1}$ are found on 24 June at the head of the filament, and on 25 June inside the evolving spiral. Afterwards, the maximum values weaken to less than 2 cm s$^{-1}$ and the ageostrophic speed reflects a multi-arm spiraliform vortex shape on 26 June. Moreover,



**Figure 4.** Top-layer (a) – (e) density $\rho$, (f) – (j) absolute horizontal density gradient $|\boldsymbol{\nabla}\rho|$, (k) – (o) frontal tendency $F$ and horizontal velocity at 2-m depth, and (p) – (t) component $Q_h$ of the frontal tendency due to horizontal advection on 24 – 28 June in R33. Vectors are drawn at 300-m resolution. All subplots are zooms and approximately centered at the local density maximum. The dashed ruler in the northwest corner of each subplot represents a total horizontal distance of 1 km.



**Figure 5.** Top-layer magnitude and direction of (a) – (e) total velocity $\boldsymbol{V}$, (f) – (j) geostrophic velocity $\boldsymbol{V}_{geo}$, (k) – (o) ageostrophic velocity $\boldsymbol{V}_{ageo}$, and (p) – (t) the ratio $|\boldsymbol{V}_{ageo}|/|\boldsymbol{V}_{geo}|$ on 24 – 28 June in R33. The color axis in (p) – (t) was limited by 100% because infinitely high ratios occur for $|\boldsymbol{V}_{geo}| = 0$. Vectors are drawn at 300-m resolution. All subplots are zooms and and approximately centered at the local density maximum. The dashed ruler in the northwest corner of each subplot represents a total horizontal distance of 1 km.




the ageostrophic flow exhibits qualitative similarities with Fig. 10h in Gula et al. (2016), particularly the divergence at the tip of the hook. The horizontal distribution of the ratio between the ageostrophic and the geostrophic speed (panels p – t) reveals that the ageostrophic portion is large in regions where the geostrophic flow is extremely weak. For instance on 24 June, $|V_{geo}|$ is close to 0 cm s$^{-1}$ in the northwest quadrant while $|V_{ageo}|$ is between about 1 cm s$^{-1}$ and the extreme value of 2.7 cm s$^{-1}$

(see above). The same is true for the spots marked in red on the following days, especially in the center of the spiral and at the eastern boundary of the image in panel q. However, ratios between 40% and 60% are found in the filament on 24 June and in the hook-shaped structure on 25 June in areas where $|V_{geo}|$ is significantly larger than zero, indicating that the roll-up is a highly nonlinear process.

## 3.3 Vorticity and strain

The formation of the spiral can be considered as the deformation of an incompressible fluid, thus it is controlled by rotation and shear strain, which are represented in the present case by the relative vorticity

$$\zeta = (\nabla \times \mathbf{V})_z = \frac{\partial v}{\partial x} - \frac{\partial u}{\partial y} \tag{2}$$

and the horizontal strain rate

$$\epsilon = \sqrt{\left(\frac{\partial u}{\partial x} - \frac{\partial v}{\partial y}\right)^2 + \left(\frac{\partial v}{\partial x} + \frac{\partial u}{\partial y}\right)^2}. \tag{3}$$

Here, $x$ and $y$ are the Cartesian coordinates pointing to the east and to the north, $z$ is the vertical coordinate, and $u, v$ are the zonal and meridional components of the total velocity vector. Furthermore, the Okubo-Weiss parameter

$$\eta = \epsilon^2 - \zeta^2 \tag{4}$$

is a relative measure indicating whether the deformation is primarily driven by relative vorticity ($\eta < 0$) or horizontal strain ($\eta > 0$).

Near the sea surface, the normalized relative vorticity $\zeta' = \zeta/f$ ($f$ is the Coriolis frequency) exhibits a maximum value of 12.3 during the coiling of the spiral on 25 June (Fig. 6b), while in the days before and after, the maximum is around 4. At the same time, the normalized anticyclonic vorticity reaches a minimum of -2.2 in an extremely narrow ribbon right to the south of the concave part of the cyclonic pattern. On the other days, it remains between -0.9 and 0. While in adiabatic flow, $\zeta' > -1$ should hold, $\zeta' < -1$ is probably a consequence of the diabatic contribution from the biharmonic diffusivity, caused by the

extreme horizontal density gradient (cf. Fig. 4g). For comparison, we recall the observations of Marmorino et al. (2018). They found maximum $\zeta'_{max} = 12.5$ in the core of a cyclone comparable in size, that is remarkably close to our value. The $f$-scaled strain rate $\epsilon' = \epsilon/f$ is also maximum on 25 June (panel g) with values > 18, while on the other days it attains values between 4 and 6. Striking is the quadrupole shape that begins to evolve on 26 June and becomes clearly visible during the successive days. Another eye-catching feature is the spot with $\epsilon' = 5.4$ close to the center of the image in panel j. Also these numbers

resemble closely the corresponding ones of Marmorino et al. (2018), $\epsilon'_{max} = 18.8$ and a mean of 6.75. The patchy pattern of $\epsilon$ in their Fig. 4F shows some agreement as well.







**Figure 6.** Top-layer (a) – (e) relative vorticity, (f) – (j) strain rate, and (k) – (o) Okubo-Weiss parameter on 24 – 28 June in R33. While the quantities in columns 1 and 2 are scaled by the Coriolis frequency $f$, the Okubo-Weiss parameter is scaled by $f^2$. All subplots are zooms and approximately centered at the local density maximum. The dashed ruler in the northwest corner of each subplot represents a total horizontal distance of 1 km.





The scaled Okubo-Weiss parameter $\eta' = \eta/f^2$ exhibits the largest spread on 25 June with $-106 < \eta' < 258$ (panel l), indicating a bimodal structure with extremely strong control of both vorticity and even more strain. Here, vorticity dominates in the center of the spiral and along the hook. Maximum strain dominated areas are also constrained to the hook, but they are absent in the center. The day before (panel k), $\eta'$ reveals distinct positive areas, hence the northward progression of the filament is

mainly controlled by deformation due to horizontal strain. However, the spread of $\eta'$ between about -3 and 17 is much smaller than in panel l. After 25 June, only the deformation in the inner part of the spiral is controlled by vorticity (panels m – o, except for the blue spot in the center of panel o) while the outskirts are dominated by strain. The corresponding numbers for $\eta'$ lie between -17 and 30. While $\eta$ was not shown in Marmorino et al. (2018), qualitative similarities arise from a comparison with Fig. 10b in Gula et al. (2016); these are the positive values at the periphery and in the outer areas and the negative numbers in

the center of their mesoscale cyclone.

### 3.4    Vertical motion

As ROMS is a hydrostatic model, the vertical velocity $w(z)$ is computed from the horizontal divergence $\delta = \partial u/\partial x + \partial v/\partial y$ by integration,

$$w(z) = \int\limits_{-H}^{z} \frac{\partial w}{\partial z} dz = -\int\limits_{-H}^{z} \delta(z) dz, \tag{5}$$

where $H$ is the water depth. In ROMS, the integral is actually computed as a sum from the bottom upwards and also as a sum from the top downwards, resulting in a linear combination of the two, weighted so that the surface down value is used near the surface while the other is used near the bottom (Hedström, 2018). Thus, the near-surface vertical velocity largely reflects the divergence pattern in the surface layer. This is shown in Fig. 7 on 24 June, when strongly convergent flow in the surface layer at the head of the filament (the blue patch in the northeast quadrant of panel a) is correlated with downwelling (panel f). The

vertical extension of the downwelling cell is 20 m, and extreme vertical velocities are found at about 8-m depth (panel k). One day later, the convergence zone becomes strained and attains the hook-like shape mentioned above. To the south of the hook appears a large area of divergent flow and associated upwelling (panels b, g, l).

     A perfect multi-arm spiral has finally developed on 26 June (panel c). Later on, the signs are mounting that the arms of the spiral disaggregate into less organized structures although the vortex is still intact. This tendency is confirmed by panels

e and j; namely, the structure of the divergence looks still spiral-like, but the vertical motion exhibits an increasingly patchy pattern (panels j, o). Concurrently, the spread of the vertical velocity at 5-m depth is increasing. While on 24 June, -45 m day$^{-1}$ $< w <$ 49 m day$^{-1}$, the spread on 26 and 27 June is -81 m day$^{-1}$ $< w <$ 60m day$^{-1}$.

     The progressive decorrelation between the surface divergence and $w$ at 5-m depth is due to sign changes of $\delta$ with depth according to equation (5); hence, convergent and divergent layers alternate. The impact of the sign changes of $\delta$ in the vertical

direction becomes evident from a comparison of the vertical structures of $w$ in panels k and o: while in panel k the vertical motion is mostly unidirectional between the surface and about 20-m depth, a sign change is evident in panel o at a depth of about 12 m. Thus, if the horizontal flow is convergent above 12 m, it is divergent below and vice versa.





**Figure 7.** (a) – (e) Top-layer $f$-scaled horizontal divergence, (f) – (j) vertical velocity $w$ at 5-m depth, and (k) – (o) zonal vertical sections of $w$ along the dashed lines in panels (a) – (j) on 24 – 28 June in R33. Subplots (a) – (j) are zooms and approximately centered at the surface density maximum. Vertical black lines are drawn in the center of each subplot. The dashed ruler in the northwest corner of the subplots (a) – (j) represents a total horizontal distance of 1 km. The rectangles in panels d, i indicate the zoomed areas in Fig. 9.





The similarity of the divergence pattern in Fig. 7c, d with Fig. 1 is striking, but a comparison of the modelled $\delta/f$ extremes (-3.0 and 1.8 on 25 June) yields little agreement with observations. For instance, the corresponding numbers from Ohlmann et al. (2017) are about -2 and 5, and $\pm12.5$ from Marmorino et al. (2018). Also, the maximum vertical speed in the model (81 m day$^{-1}$) is much smaller than the estimates of 240 m day$^{-1}$ and $\mathcal{O}(1000$ m day$^{-1})$ obtained by Ohlmann et al. (2017)

and D'Asaro et al. (2018), respectively. In contrast, the radial structure of $\delta$ in Fig. 6C of Marmorino et al. (2018) resembles closely the oscillations along the dashed line in Fig. 7c which cuts across several streaks indicating alternately convergence and divergence due to the spiraliform divergence pattern. The corresponding patterns of $w$ were reported in several studies of mesoscale anticyclones, e.g. in Koszalka et al. (2009) and Estrada-Allis et al. (2019), while in other articles, dipolar or multipolar structures were found (Adams et al., 2017; Barceló-Llull et al., 2017; Buongiorno Nardelli, 2013).

## 3.5   Vertical stratification

The evolution of the vertical stratification within the spiral and in its immediate neighborhood is displayed by means of the mixed-layer depth MLD, the depth $h_\rho$ of the $\rho = 5.3$ kg m$^{-3}$ isopycnal, and vertical density sections through the eddy center shown in Fig. 8. For the determination of the MLD, a $\Delta\rho = 0.1$ kg m$^{-3}$ criterion was applied; thus the MLD is the depth where the in-situ density exceeds the surface density by more than 0.1 kg m$^{-3}$ for the first time. The overall deepest MLD >

16 m is located in the center of the filament and the shallowest < 3 m in the southwest quadrant of panel a, both on 24 June. The deep and shallow areas lie side by side and are organized in a southwest-northeast oriented finger-like pattern, exhibiting MLD gradients on the order of 10 m over a horizontal distance of 100 m. During the following 24 hours, the strong gradients are retained, but the maximum MLD decreases to $\approx 11$ m on 25 June on the western flank of the coiling spiral (panel b). A further decrease of the maximum MLD to about 9 m and consecutive weakening of the gradients occurs until 28 June (panels

c – d), when the difference of the MLD in the center of the eddy and the periphery is less than 4 m. Noteworthy is also the shape of the MLD contours. While on 25 June the contours are reasonably smooth, more irregular asymmetric patterns prevail thereafter.

On 24 June (Fig. 8f), the minimum of $h_\rho$ is not defined because the isopycnal outcrops at the sea surface. On 25 June, however, it is completely subducted, and the minimum $h_\rho$ is slightly less than 3 m. The values of the minima on the successive

days indicate that the isopycnal is slowly sinking in the center of the spiral, attaining a depth of about 3.6 m on 28 June while at the outskirts it rises by about 1 m. Similarly to the MLD, the shape of the $h_\rho$ contours becomes irregular and exhibits a wavy pattern after 25 June. The density cross sections of 24 to 28 June (panels k – o) depict clearly the doming of isopycnals in the center of the spiral, the surface outcrop of the $\rho = 5.3$ kg m$^{-3}$ isopycnal and its subsequent subduction and sinking. While there is only one dome in panels k and m – o, a second dome appears on 25 June at a distance of about 4000 m where the

section cuts through the umbilical. Furthermore, evidence is provided for the leveling of the MLD and the wavy patterns of $h_\rho$ that are most pronounced in panels l and m.

The subduction of the 5.3 kg m$^{-3}$ isopycnal in the center, the contemporaneous rising at the periphery of the spiral, and the leveling of the MLD are clear indicators for restratification by mixed-layer instability (Boccaletti et al., 2007; Fox-Kemper and







**Figure 8.** (a) – (e) Mixed-layer depth from a $\Delta\rho$ =0.1 kg m$^{-3}$ criterion, (f) – (j) depth of the $\rho$ =5.3 kg m$^{-3}$ density surface, and (k) – (o) zonal vertical sections of density along the dashed lines in (a) – (j) on 24 – 28 June in R33. Subplots (a) – (j) are zooms and approximately centered at the surface density maximum. White black lines are drawn in the center of each subplot. The dashed ruler in the northwest corner of (a) – (j) represents a total horizontal distance of 1 km.





Ferrari, 2008), supporting the forward energy cascade in the ocean (McWilliams, 2019). A closer examination of the irregular MLD patterns and the wavy patterns of $h_\rho$ after 25 June is accomplished in the following subsection.

## 3.6 Vortex Rossby waves

The above investigations have shown that the fully developed cyclone is not axisymmetric. While the density pattern (e.g. Fig.

4d) is ellipsoidal, many quantities are spiraliform, and other quantities such as the horizontal velocity, MLD, and $h_\rho$ exhibit azimuthally irregular or wavy patterns. Similar non-axisymmetric structures, e.g. spiral-shaped rain or cloud bands, are also observed in atmospheric cyclones. According to various theoretical studies (Montgomery and Kallenbach, 1997; Chen and Yau, 2001; Wang, 2002a, b), they are caused by vortex Rossby waves (VRWs). Unlike planetary Rossby waves, the restoring force of which is the meridional gradient of the Coriolis frequency, VRWs owe their existence to the radial and azimuthal gradients

of the absolute vorticity. While the radial gradient enables their outward propagation, the orbital progression is driven by the azimuthal gradient, and the common action of both gradients generates the spiraliform patterns.

In the ocean, VRWs were reported for the first time by Buongiorno Nardelli (2013) who described the evolution of a mesoscale cyclonic eddy by means of observations and inviscid, adiabatic semigeostrophic equations. The synthetic estimates show that the vertical velocity field in the eddy interior is dominated by azimuthal oscillations due to VRWs. Complex multi-

polar vertical motion patterns in anticyclonic mesoscale eddies were diagnosed by Barceló-Llull et al. (2017) and Estrada-Allis et al. (2019). In agreement with the papers of Viúdez and Dritschel (2004) and Pallàs Sanz and Viúdez (2005), the horizontal advection of relative vorticity was found to be the main driver for the vertical motion described in both publications.

In the present case, VRWs attracted our attention due to the wavy $h_\rho$ pattern in Fig. 8h, i, j. The most striking example was found on 26 June at 22h, where the radial excursions of the $h_\rho = 5$-m contour attained a maximum of about 300 m (Fig. 9a).

The wavelength of the oscillations on all $h_\rho$-contours is on the order of 100 m, and from the animation of Onken et al. (2021a), a rather crude estimate of the cyclonic angular velocity arrived at $4 \times 10^{-5}$ rad s$^{-1}$, equivalent to an orbital period around 40 hours. According to the findings above, the vorticity advection $ADV = \boldsymbol{V} \cdot \nabla \zeta^a$ is the main driver of the vertical motion in eddies, where $\zeta^a = \zeta + f$ is the absolute vorticity. The components of $ADV$ at 2-m depth are displayed in panels b and c. The azimuthal oscillations of the $h_\rho$-contours are reflected by the horizontal speed $|\boldsymbol{V}|$ in panel b. Thus, water particles circulating

the cyclone accelerate and decelerate alternately. In panel c, $\zeta^a/f$ exhibits a rather complex pattern. The initial spiral (cf. Fig. 6) is still discernable, but in the eddy center emerged anomalous structures. The $ADV$ structure (panel d) resembles almost perfectly the top-layer divergence and the corresponding vertical motion patterns in Fig. 7d and i, which is two hours later. Instead of $ADV$, we have plotted $-ADV$ in order to match the colors. Hence, positive vorticity advection (blue) correlates with convergent flow and downwelling, while $ADV < 0$ (red) indicates divergence and upwelling.

The most important message of Fig. 9 is that the spiral structure of the vertical motion pattern is caused by VRWs. If there were no VRWs, the eddy would be circular and no non-zero vorticity transport would be accomplished, because the vectors of the horizontal velocity were parallel to contours of $\zeta^a$. Such a situation is approximately met in the southwest quadrant of panel c. Consequently, the spiral arms there are less pronounced.



**Figure 9.** Properties of vortex Rossby waves on 26 June 22:00h, (a) depth $h_\rho$ of the isopycnal $\rho = 25.3$ kg m$^{-3}$, (b) magnitude and direction of horizontal velocity $\boldsymbol{V}$, (c) absolute vorticity $\zeta_a/f$ with horizontal velocity vectors superimposed, and (d) negative vorticity advection $-\boldsymbol{V} \cdot \boldsymbol{\nabla}\zeta_a$ at 2-m depth. The dashed bars represents a horizontal distance of 500 m. For the position of the zoomed area see Figs. 7d, i.





### 3.7 Secondary instabilities

The objective for the setup of R33 was to provide a detailed description of a cyclonic spiral that was generated in the coarser resolution parent model R100. As the diameter of the spiral was about 1 km and no smaller spirals were found in R100, it was speculated that the minimum size of circular eddies in any circulation model was about 10 times the grid size. This is

in agreement with the grandparent model R500 which did not generate any eddies smaller than about 5 km (Onken et al., 2020a). Consequently if this rule holds, R33 should be able to create eddies even smaller than 1 km. And in fact, several of such mini eddies were detected in the R33 model run. The most prominent example is exhibited in Fig. 10: it illustrates the birth of 3 small submesoscale cyclonic spirals by means of the absolute horizontal density gradient in the wake of the spiral above. The evolution starts on 26 June around noon (panel a) with a small perturbation of the density front. Within 12

hours, the perturbation has amplified and generated the first spiral with a diameter of around 500 m. At the same time, another perturbation starts to grow upstream. Half a day later (panel c), two additional meanders emerge downstream, leading to the formation of a train of 3 spirals on the whole.

A similar situation is depicted in Fig. 1a of Callies et al. (2015) where secondary instabilities with wavelengths around 20 km grow at the perimeter of a cyclonic Gulf Stream eddy of about 80 km in diameter (This figure originates from the numerical

model of Gula et al. (2015) with 750-m horizontal resolution; see also Callies (2016)). Moreover, in the framework of the "Expedition Clockwork Ocean" (https://uhrwerk-ozean.de/, last access 10 November 2020), the evolution of an even smaller eddy of about 300 m in diameter was observed in the Baltic Sea (Fig. 11).

### 4 Drifters and floats

As atmospheric forcing is turned off, the dynamics of R33 can be considered as adiabatic and nearly frictionless because no

explicit vertical mixing is specified, and the biharmonic diffusivity coefficient is extremely small. Hence, for any material water particle,

$$dq \equiv 0. \tag{6}$$

where

$$q = \frac{1}{\rho_0} \frac{\zeta_\rho^a}{H} \tag{7}$$

is the potential vorticity expressed in isopycnal coordinates (Müller, 1995). Here, $\rho_0$ is the mean density of the respective isopycnal layer, $\zeta_\rho^a = \zeta_\rho + f$ the absolute vorticity, and $\zeta_\rho$ the relative vorticity evaluated along a surface of constant density. $H$ is the vertical spacing between the shallower isopycnal with density $\rho_0 - \Delta\rho/2$ and the deeper one with density $\rho_0 + \Delta\rho/2$, where $\Delta\rho$ is the density difference between the deeper and the shallower isopycnal. Because the R33 domain is rather small with about $20 \times 20 \text{ km}^2$, $f$ is considered to be constant. Hence, material changes of the absolute vorticity are compensated by

changes of the layer thickness and vice versa:

$$\frac{d(\zeta_\rho^a)}{\zeta_\rho^a} = \frac{dH}{H} \tag{8}$$




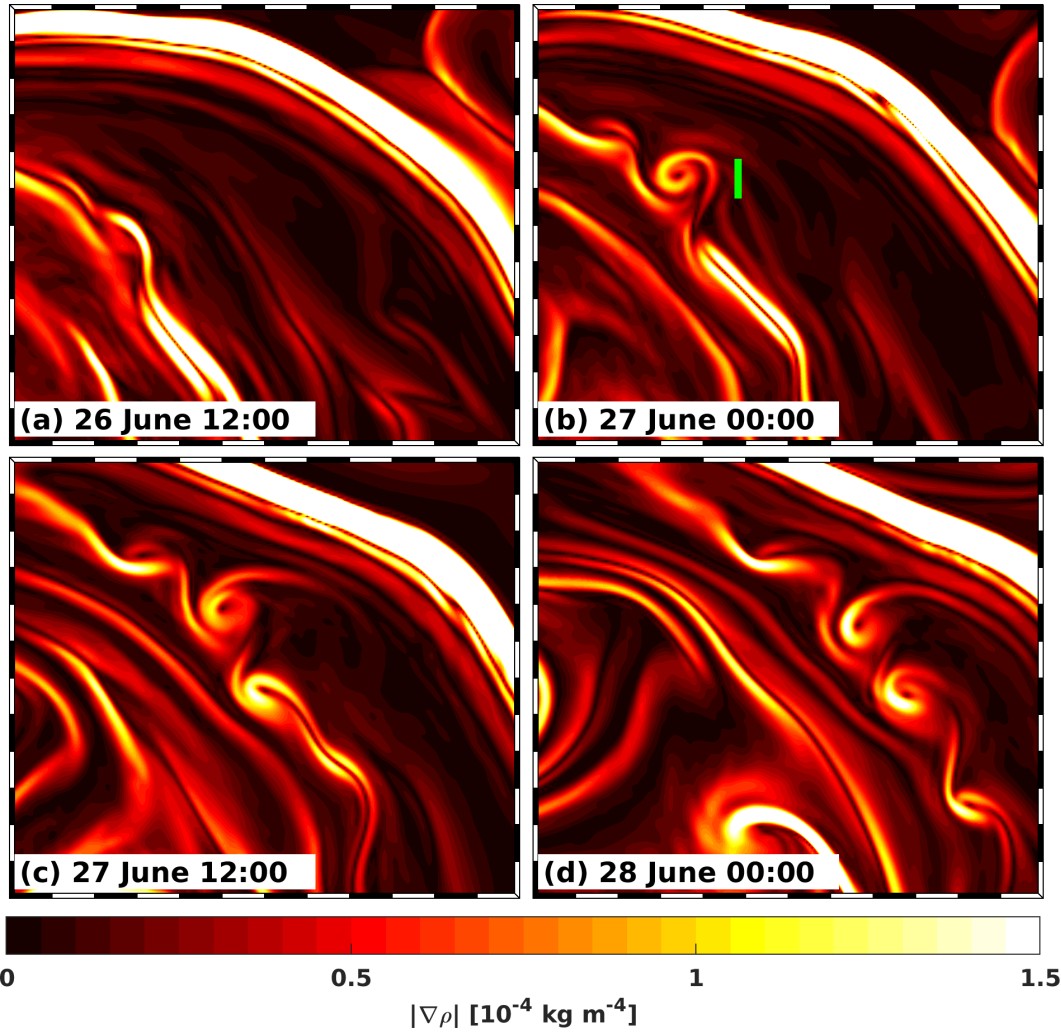

**Figure 10.** Zoomed snapshots in 12-hour intervals of the absolute horizontal density gradient in the top layer exhibit the evolution of a train of secondary instabilities in the wake of a cyclonic spiral. The position of the image section is marked by the white rectangle in Fig. 3j. The green bar in panel b represents a horizontal distance of 500 m.

For instance, the height of a water column increases, if the vorticity increases (vortex stretching), and a decrease of the vorticity causes shrinking of the water column (vortex squeezing). In the northern hemisphere and in the absence of diabatic processes, $q$ is always positive (Harvey, 2020), i.e. $\zeta_\rho^a >= 0$. Consequently, anticyclonic relative vorticity cannot be smaller than the negative Coriolis parameter while the values of cyclonic vorticity are not constrained. An important consequence of potential vorticity conservation is that material changes of any kinematic or dynamical quantity are intimately and mutually related to material changes of $\zeta_\rho^a$ or $H$. For example, convergent flow ($\delta < 0$) is a consequence of vortex stretching due to increasing vorticity or vice-versa, while $\delta > 0$ is correlated with decreasing vorticity.







**Figure 11.** Sea surface temperature of a very small cyclonic eddy observed on 27 June 2016 during the "Expedition Clockwork Ocean" south of Bornholm (cf. Fig. 2). The image was taken with a infrared camera onboard a zeppelin which "parked" over the eddy and recorded its life cycle.





**Table 1.** Setup of experiments with drifters and floats. ($NN$: amount of drifters/floats, $t_0$: deployment time, $D$: deployment depth, $x_W$: western, $x_E$: eastern, $y_S$: southern, $y_N$: northern boundary)

| Name | $NN$ | $t_0$ | $D$ [m] | Distribution | $x_W$ | $x_E$ | $y_S$ | $y_N$ |
|---|---|---|---|---|---|---|---|---|
| | | | | Experiments with isobaric drifters | | | | |
| ISOBAR1 | 134 | 24 June 00:00 h | 1 | zonal line, equidistant | 14°36.00' E | 14°44.00' E | 54°37.38' N | 54°37.38' N |
| ISOBAR2 | 100,000 | 24 June 00:00 h | 1 | area, random | 14°32.00' E | 14°46.00' E | 54°35.00' N | 54°42.00' N |
| | | | | Experiments with isopycnal floats | | | | |
| ISOPYC1 | 134 | 24 June 00:00 h | 1 | zonal line, equidistant | 14°36.00' E | 14°44.00' E | 54°37.38' N | 54°37.38' N |
| ISOPYC2 | 100,000 | 24 June 00:00 h | 1 | area, random | 14°32.00' E | 14°46.00' E | 54°35.00' N | 54°42.00' N |
| ISOPYC3 | 100,000 | 24 June 00:00 h | 10 | area, random | 14°32.00' E | 14°46.00' E | 54°35.00' N | 54°42.00' N |
| ISOPYC4 | 100,000 | 25 June 00:00 h | 1 | area, random | 14°32.00' E | 14°46.00' E | 54°35.00' N | 54°42.00' N |

In the following and in order to explore and understand some of its properties, the cyclonic spiral is larded with isopycnal floats which are assumed to represent material (or Lagrangian) water particles conserving potential vorticity. Before, however, two precursor experiments are conducted with isobaric floats that are launched close to the sea surface. Namely, such floats do not conserve potential vorticity but it is expected that they behave like real surface drifters; therefore, they are denoted as isobaric drifters or just "drifters".

## 4.1 Isobaric drifters

In the first experiment, named ISOBAR1 (see Table 1), 134 isobaric drifters were launched on 24 June at 00:00 h along 54° 37.38' N, and equally spaced at 6.5 m between 14° 36.0' E and 14° 44.0' E (Fig. 12a). The position of the zonal line was selected in a way that it intersects the density maximum of the filament which becomes the center of the cyclonic spiral during the formation process. The deployment depth of the drifters was $D$=1 m below the sea surface, e.g. at $z \approx -0.85$ m in the Cartesian space, because the sea surface height in the entire domain is around 0.15 m. The drifters were sequentially numbered from west to east, starting at #1. While the initial positions of drifters #54 and #55 were right in the density maximum, drifters #1 – #53 were located to the west and #56 – #134 to the east of the maximum (see subpanel a).

By 26 June (panel f), 48 drifters were captured by the orbital motion of the spiral and completed at least one circuit. In detail, these are

- #51 – #53 to the west of the initial density maximum (5.29 kg m$^{-3}$ < $\rho$ < 5.32 kg m$^{-3}$),

- #54 and #55 in the center of the density maximum ($\rho$ > 5.32 kg m$^{-3}$),

- #56 – #98 to the east of the density maximum (5.16 kg m$^{-3}$ < $\rho$ < 5.32 kg m$^{-3}$).





**Figure 12.** Experiment ISOBAR1: (a) – (i) Gray and magenta dots denote the positions of 134 drifters between 24 June 00:00 h and 28 June 00:00 h in 12-hour intervals within a zoomed area of the R33 domain (cf. Fig. 3). Drifters that were captured by the orbital motion of the spiral are marked by magenta dots. The tails appended to the dots represent the track of each individual drifter during the previous 3 hours. The drifter positions are superimposed to the top-layer density, the color map is the same as in Fig. 3. The drifter numbers in intervals of 10 are indicated in subpanel (a). Drifters #70 and #90 are encircled black and white, respectively.





The (ambient) density of the 48 drifters over the course of the model run is shown in Fig. 13a in a contour plot. The initial density at 0 hours reflects the above description with just one density maximum for #54 and #55, and minima for #51 and #98. At the end of the model run at 96 hours, the arrangement has changed: now, there are two density maxima centered at about #57 and #85, two minima for #51 an #98, and a third minimum centered at about #70. That means that some drifters or drifter

clusters swapt their position relative to the absolute density maximum. By way of example, we compare the histories of #70 and #90: as at 0 hours, the density of #70 was higher than the density of #90, #70 was closer to the high-density center of the filament. In contrast at 96 hours, the density of #90 is higher than that of #70, thus #90 is closer to the center of the spiral. This "transformation phase" took place during the early stages of the eddy formation between about 0 hours and 36 hours, where the density of all drifter changes. The corresponding swapping of positions is illustrated by Fig. 12 where #70 and #90 are

encircled black and white, respectively. On 25 June at 00:00h, #90 is situated at the northern tip of the filament while #70 has completed a first orbit around the cyclone and lies a few hundred meters south. At noon of the same day, #70 finalized another orbit and #90 its first one, but now #70 is located north of #90 on an outer track and its density is lower than that of #90. Thus, the swapping occurred in the early morning of 25 June when #70 crossed the track of #90.

The coiling of the filament is accomplished by horizontal strain and divergent motion, that attain extreme values on 25 June

with a strain rate of $18f$ and a divergence of $-3f$. As can be seen from Figs. 6g and 7b, the positions of these extremes coincide with the positions of drifters #51 – #98, which are finally trapped inside the eddy. Half a day before, these drifters were arranged along two separate sections of a continuous line, but the large strain rate together with the convergence merges those drifters onto a single line, that is spooled into the eddy. The entire process is similar to the zipper structure of real surface drifters as described by D'Asaro et al. (2018). At the same time, the heavy water subducts due to the lateral advection of lighter water by

the convergent flow (see Figs. 8k, l). The latter is clearly illustrated by Fig. 13b: on 24 June between 6:00 h and 18:00 h, most drifters are exposed to strong convergences and the water at the drifter position becomes lighter. Later on, mostly divergent flow pushes the drifters again into heavier water where they remain until the end of the integration. However, this is not true for drifters #67 – #71: after about 30 hours, they are forced by another convergence into lighter water. After 36 hours, the density adjustment of all drifters calms down but one notices still smaller density fluctuations until the end of the model run.

These fluctuation are accompanied by sign changes of the divergence along the drifter tracks (cf. Fig. 7c, d, e) that cause lateral oscillations of the drifters across isopycnals. While the latter are caused by ageostrophic velocities and are oriented radially with respect to the eddy center, fluctuations of the drifter speed caused mainly by geostrophic velocities are oriented in the azimuthal direction as shown in Figs. 13c and Fig. 5b–e. Hence, the drifters are alternately accelerated and decelerated along their trajectory as already descrbed in Section 3.6.

The above experiment included only 134 drifters – it was therefore easy to identify the ones that were captured by the circular circulation of the eddy and to describe the individual properties of the ambient fluid along their track. However, the quantity of drifters was not sufficient in order to simulate any observed tracer pattern at the sea surface. Therefore, we repeated the experiment with 100,000 drifters, that were randomly distributed in a rectangular area on 24 June (Fig. 14b). This experiment is denoted ISOBAR2 (see again Table 1) . Two days later on later on 26 June (Fig. 14a), the drifters became arranged in a

spiraliform pattern, similar to $|\nabla\rho|$ and $\zeta/f$ in Figs. 4h and 6c, respectively. Hence, the high-concentration bands resemble the







**Figure 13.** Experiment ISOBAR1: Along track (a) density, (b) horizontal divergence normalized by the Coriolis frequency $f$, and (c) horizontal speed of drifters #51 – #98 that were caught by the orbital motion of the spiral. Vertical dashed lines indicate the instants shown in Fig. 12.





single-arm shapes of the surface density front and the vorticity, but they are different from the pattern of the multi-arm shaped horizontal divergence shown in Fig. 7c. In conclusion, the spiral structure of the cyanobacteria in Fig. 1 was certainly generated by alternating convergent and divergent currents but it may not reflect the instantaneous divergence.

On may argue that the behavior of the drifters is unrealistic because the density is modified along their path. Indeed, this
must not happen with Lagrangian floats representing material water particles, but as isobaric drifters are forced to stay at a predefined pressure level, they cannot behave like material particles. For instance, in convergent flow, density is merely an Eulerian environmental parameter for isobaric drifters that changes in response to the subduction process, while Lagrangian (isopycnal) floats subside in order to preserve their material property "density". Amongst others, this different behavior is demonstrated in the following subsection.

## 4.2 Isopycnal floats

An additional experiment is carried out with 134 isopycnal floats (ISOPYC1, see Table 1). The floats are launched exactly at the same positions and at the same time as in the experiment ISOBAR1 with the isobaric drifters. The horizontal positions of the floats are displayed in Fig. 15. After 12 hours (panel b), they closely resemble the corresponding positions of the isobaric drifters (cf. Fig. 12b), but already on 25 June (panels c), the floats trapped by the cyclonic spiral cover a larger area than the
corresponding isobaric drifters. This comparison is legitimate because the same floats #51 – #98 plus #99 and #100 participate in the early stage of the coiling, using the same numbering as in the experiment ISOBAR1. However, the behavior of #53 – #55 (black dots) is rather remarkable: instead of performing orbital motions around the center of the spiral, they veer immediately away from the cyclone into the ambient water and continue traveling in southwesterly direction (panels c–f). Surprisingly, the initial positions of these runaways were located inside the density maximum of the filament (panel a) that became the center of
the spiral. Later on during the course of the integration, the remaining floats stay inside the cyclone.

Time series of the density of floats #51 – #100 (Fig. 16a) reveal that this quantity is reasonably conserved for all floats during the entire model run, except for #53, the final density of which on 28 June is 0.09 kg m$^{-3}$ lower than the initial density. During the first 48 hours, all floats sink to greater depth (Fig. 16b). Thereafter, many of them raise again towards the surface, but others stay at depth or oscillate vertically. By consideration of the maximum depths, the floats may be separated into 3 groups:

– Group I floats (magenta dots in Fig. 15) descend to maximum depths of about 5 m; the 24 members of this group are #58 – #74 and #94 – #100. According to Fig. 15a, #58 – #74 originate from the inner part of the filament and descend to a maximum depth of 4–5 m between about hours 12 and 24; most of them return to shallower levels thereafter. In contrast, the initial positions of #94 – #100 are about 2 km to the east of the center of the filament; they descend almost constantly to their final depth between 3 m and 4 m.

– Group II floats (orange) descend to maximum depths between 5 and 10 m; the 23 member of this group are #51, #52, #56, #57, and #75 – #93; their initial positions were located very close to the the density maximum of the filament and between about 1 and 2 km east of the maximum.

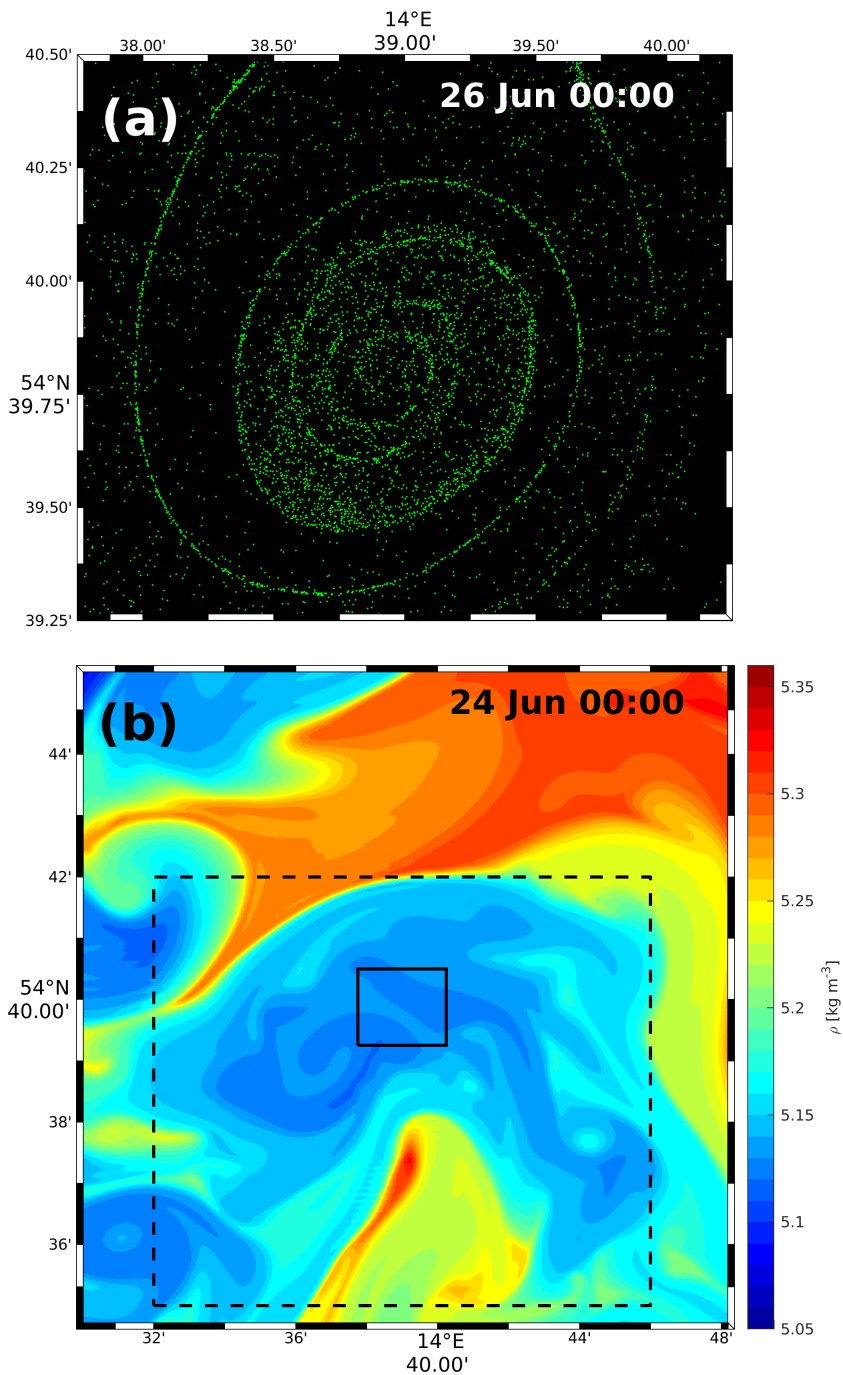

**Figure 14.** Experiment ISOBAR2: (a) Positions of drifters on 26 June within the black frame shown in frame b, (b) 100,000 drifters were distributed randomly within the dashed rectangle on 24 June.





**Figure 15.** Experiment ISOPYC1: (a) – (i) Gray and magenta dots denote the positions of 134 floats between 24 June 00:00 h and 28 June 00:00 h in 12-hour intervals within a zoomed area of the R33 domain (cf. Fig. 3). Floats that were captured by the orbital motion of the spiral are marked by magenta (Group I), orange (Group II), and black (Group III) dots. For the float groups, see text. The tails appended to the dots represent the track of each individual float during the previous 3 hours. The float positions are superimposed to the top-layer density, the color map is the same as in Fig. 3.




– Group III floats (black) descend to maximum between 12 m and 15 m; the 3 members of this group are the runaways #53 – #55.

Using the example of float #64, some more light is shed on the complex vertical motion pattern. This float was selected, because it exhibits a rather unsophisticated behavior. According to Fig. 16a, it preserves its density of about $\rho_0 = 5.243$ kg

m$^{-3}$ during the entire course of the model integration, and major depth changes occur only twice: between hours 0 and about 12, this float sinks from the initial vertical position to about 4 m, and thereafter it climbs rapidly to about 2-m depth and remains almost constantly at this level after about 30 hours (panel b). This low-frequency vertical oscillation is apparently a consequence of the subduction of the corresponding isopycnals as illustrated by Fig. 8f, g, k, l, and a subsequent equilibration. Later than hour 30, the float performs yo-yo like high-frequency vertical oscillations with small amplitude according to panel

c. We postulate that these oscillations are driven by material changes of the vorticity that is compensated by changes of the layer thickness according to equation (8). The latter in turn requires non-zero divergence and associated vertical velocities as specified by equation (5). In order to demonstrate the interplay of these quantities, we performed an OSSE-like (Halliwell et al., 2017) simulation experiment, where the float trajectory stands for the observations and the ROMS output represents the environment of the float, i.e. the truth. As the float is isopycnal, we projected in the first step the prognostic variables of the

ROMS output on the 5.243 kg m$^{-3}$ isopycnal and computed $\zeta_\rho^a$, $H$, and $q$ using $\Delta\rho = 0.01$ kg m$^{-3}$ (see equations (7) and (8)). The result is displayed in Fig. 17. The time series in panels a, b, c are created from the trajectory data which are available at each model time step of 20 s. In contrast, the graphs in panels d, e, f are extracted from the ROMS output. As the output is only available in intervals of 6 minutes, $\zeta_\rho^a$, $H$, and $q$ were interpolated in time and along the respective isopycnal in space at the exact horizontal position of the float. Fig. 17b shows the vertical position of the float. Altough this plot contains the same

information as Fig. 16b, it exhibits high-frequency oscillations that are merely visible in the contour plot. These oscillations are even more pronounced in the graph of the vertical velocity $w$ (panel c), which was computed from the time derivative of $z$. Rapid oscillations are also characteristic for $\zeta_\rho^a$ and $H$ (panel d, e) for times later than 27 hours. Before, there is no clear signal detectable, because the isopycnal $\rho_0$ outcrops intermittently at the sea surface. The critical question arises whether $\zeta_\rho^a$ and $H$ are related to each other. And indeed, a correlation analysis yields a significant coefficient of 0.64; hence, fractional

changes of the absolute vorticity equal largely fractional changes of the layer thickness as required by equation (8). Ideally, $q$ should be individually conserved, but according to panel f, it fluctuates between about $0.6 \times 10^{-3}$ m$^{-1}$ s$^{-1}$ and $1.3 \times 10^{-3}$ m$^{-1}$ s$^{-1}$. Moreover, $\zeta_\rho^a$ and $w$ are not at all correlated, in agreement with Buongiorno Nardelli (2013). Namely, there is a direct relationship between $\zeta_\rho^a$, $H$ and the divergence $\delta$, however, $w$ does not depend on $\delta$ but on its vertical integral (equation (5)). What has been said above implies that material vorticity changes drive the vertical motion. By way of example, Fig. 18

illustrates how such changes may come about. It shows the positions of float #64 at four different instants in one-hour intervals on 26 June, together with a map of the absolute isopycnal vorticity. At 9:00 h (panel a), the vorticity at the position of the float is around 3, at 10:00 h it is greater than 4, and thereafter it decreases again to about 3 at 11:00 h and less than 3 at noon. This sequence is reflected by the gray-shaded patch in Fig. 17d. Hence, the material vorticity changes are due to intersections of the float track with vorticity contours. As the float track is a proxy for the horizontal velocity, this is equivalent to non-zero

vorticity advection, the same mechanism that drives VRWs (cf. Figs. 6c, d and Fig. 9).





**Figure 16.** Experiment ISOPYC1: Along track (a) density, (b) depth and (c) vertical velocity of floats #51 – #99 that were initially caught by the orbital motion of the spiral. Vertical dashed lines indicate the instants shown in Fig. 15.







**Figure 17.** Experiment ISOPYC1: Time series of material properties of float #64, (a) density $\rho$, (b) depth $z$, (c) vertical velocity $w$, (d) $f$-scaled absolute isopycnal vorticity $\zeta_\rho^a/f$, (e) layer thickness $H$, and (f) isopycnal potential vorticity $q$. The gray-shaded patch in (d) marks the time span covered by Fig. 18(a) – (d).







**Figure 18.** Experiment ISOPYC1: Horizontal positions of floats #64 (black dot) and #57 (blue; for the meaning see Fig. 19) in the cyclonic spiral on 26 June between 09:00 h and 12:00 h. The tails appended to the dots represent the tracks of each float during the previous 2 hours. The tracks are superimposed to a map of the $f$-scaled isopycnal absolute vorticity $\zeta_\rho^a / f$ along the $\rho_0$=5.243 kg m$^{-3}$ isopycnal surface. The outcrop region of that isopycnal is left white, the color map is the same as in Fig. 9c. The dashed ruler in the northwest corner of each subplot represents a horizontal distance of 200 m.





For further illustration of the kinematics during the roll-up process of the spiral, nine cutouts of an animation of the float tracks together with the shape of an isopycnal surface (Onken et al., 2021b) are lumped together in Fig. 19. The isopycnal $\rho = 25.3$ kg m$^{-3}$ is shown for two reasons: (i) this surface outcrops at the sea surface at the start of the integration on 24 June and subducts thereafter. The subduction process is finished on 24 June at 18:00 and the isopycnal remains at a depth of about

3–4 m in the center of the spiral (Fig. 8, center and right column). (ii) The majority of floats that get caught by the spiral (except for #51 – #57, cf. Fig. 16a) remain above that isopycnal during the first 36 hours. In the morning of 24 June, the isopycnal is shaped like a slightly curved mountain ridge. The curvature radius decreases rapidly and a spiraliform ridge starts to form in the evening of the same day. The spiral is clearly visible on 25 June and explains the generation mechanism of the wavelike patterns on the flanks of the isopycnal in Fig. 8. Later on, the height of the ridge flattens out, but groovings around the doming

isopycnal in the center of the eddy are still visible. The groovings are VRWs, and they circulate slowly anticlockwise around the central dome which much smaller angular speed than the angular speed of the floats. As the floats are isopycnal, they have to respond to the changing depth of the isopycnal along their path by vertical motion. This is in agreement with Buongiorno Nardelli (2013) and clearly visible in the above mentioned animation and the wavelike tails of the floats in Fig. 19.

The above experiment has shown that the dynamics of an evolving spiral leads to an export of Lagrangian particles from

the near-surface layers to greater depths. Apparently, the vertical displacement of the particles depends on the initial radial distance from the density maximum of the filament, but it is not a monotonic function of the distance, because both Group I and Group II particles are organized each in 2 modes, where the first mode is closer to the density maximum and the second is farther away. The modes of these groups are alternately arranged, and it is conjectured that the associated vertical displacements in the spiral exhibit a banded circular structure. However, as this is difficult to demonstrate with just 50 floats, we repeated

the experiment with 100,000 floats in order to obtain a statistically robust outcome (ISOPYC2, see Table 1). In Fig. 20a, the horizontal positions of floats on 26 June are marked by dots. Their color code indicates the attained depths at this particular time. Similar to the experiment with 100,000 isobaric drifters (cf. Fig. 14), the single-arm spiral of the cyclone is visible by means of the elevated distribution density of floats. While the dark and bright blue colors in the center of the spiral and adjacent to the wrapped arm suggest attained depths in the 0–5-m range, the light blue and green dots in the arm denote greater depth of

up to about 12 m. This is an indication for enhanced downwelling in the spiral arm and a radially banded structure of vertical motions. Fig. 20a also provides clear evidence for the existence of 287 runaway floats that subducted in the center of the spiral and thereafter escaped from the orbiting motion in the thermocline. The runaways are highlighted by green to red colors and attain maximum depths of up to 18 m. Fig. 20b proves that these floats started in the center of the filament on 24 June, because their mean density at the deployment depth was $5.304 \pm 0.010$ kg m$^{-3}$ (panel c). At that time, such high densities were only

present in the filament and north of 54°42'N, but not any float was deployed in the latter region (see Fig. 14a).

In order to explore whether there is also isopycnal transport in the opposite direction (upwelling), ISOPYC2 was repeated, but now, the initial depths of the floats were set to 10 m below the sea surface instead of 1 m (ISOPYC3, see Table 1). In Fig. 21a, the float positions are again represented by colored dots, but the color scale was defined in a way that it indicates the depth anomaly of each float from the initial position on 24 June: upwelling floats are marked blue, downwelling floats are

red. East of 14°39' E and in the center of the spiral, the predominant color of the dots is light-blue, indicating that these floats





**Figure 19.** Experiment ISOPYC1: Three-dimensional view of the tracks of 134 floats between 24 June and 28 June in 12-hour intervals within a zoomed area of the R33 domain. The actual float positions are marked by bold dots, the appended tails represent the track of each individual float during the previous 3 hours. The tracks are superimposed to the $\rho = 25.3$ kg m$^{-3}$ isopycnal surface and only the floats above that density surface are visible. The azimuth and the elevation of the view angle are 170° and 60°, respectively. The horizontal coordinate system is centered around float #57, that resides always in the density maximum of the cyclonic spiral (cf. Fig. 18). The full animation is shown in Onken et al. (2021b).

rose by a few meters. According to panel b, most of them were entrained from the region west of the filament, but a significant amount was also absorbed by the coiling filament while proceeding northward. The majority of sinking floats is found west of 14°39' E. These floats originate from the center of the filament, similar to the runaways in ISOPYC2. Hence, there exists apparently a mechanism close to the center of the evolving spiral that procures an efficient export of particles to greater depth, 5 and that finally enables them to escape from the vortex. The objective of the final experiment, ISOPYC4 (see Table 1), was to determine whether that mechanism is a permanent feature of the spiral. Therefore, 100,000 floats were deployed 1 day later than in all previous experiments on 25 June, again at 1-m depth and at the same positions as in ISOPYC2. Namely, there were





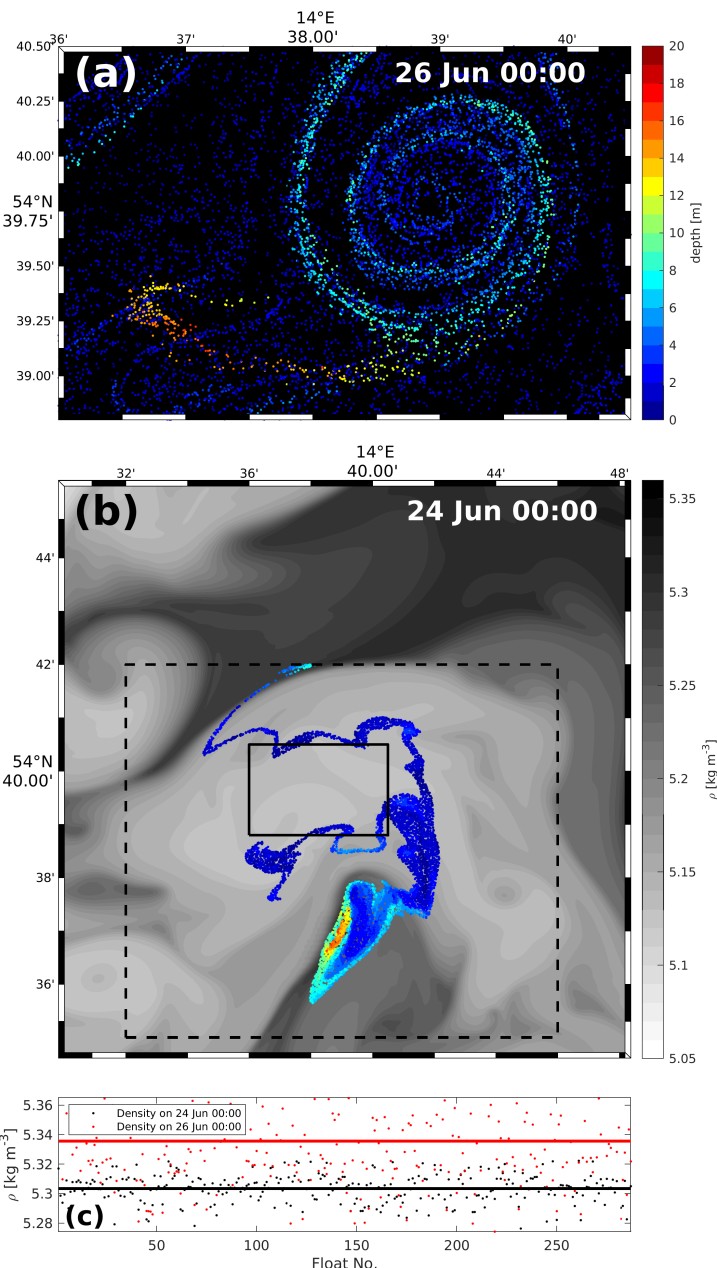

**Figure 20.** Experiment ISOPYC2: (a) Horizontal and color-coded vertical positions of floats on 26 June within the black frame shown in panel b, (b) 100,000 floats were randomly deployed at 1-m depth within the dashed rectangle on 24 June. The colored dots indicate the horizontal positions of the floats shown in panel a, using the same color code. (c) ambient density on 24 (black) and 26 (red) June of 287 floats that descended to more than 10-m depth on 26 June. The solid lines indicate the mean densities.





165 floats that sank to maximum depth between 6 m and 8 m. However, the sinking occurred along the spiral arm, only. Hence, the intense downwelling is restricted to the early formation phase of the spiral.

## 4.3 Dispersion

We define the dispersion of a cluster of N floats as

$$\Lambda(t) = \frac{1}{N} \sum_{1}^{N} |X_i(t) - \overline{X}(t)|, \tag{9}$$

where $\overline{X}(t)$ is the mean position of all floats, and $X_i(t)$ is the position of the i-th float at time $t$. Hence, $\Lambda$ is a measure of the mean distance of all floats from their "center of gravity". $\Lambda$ was computed for the model runs with isobaric (ISOBAR1) and shallow isopycnal floats (ISOPYC1); the results are shown in Fig. 22. The dashed lines refer to the dispersion of all floats, while the solid graphs show the dispersion of only those ones that were captured by the spiral. This separation was, however, only feasible for the experiments with 134 floats. The red dashed curve increases steadily during the first 33 hours, then it decreases silghtly until 44 hours, and finally increases again up to 5 km at the end of the model run. The blue dashed curve is almost identical to the red one, except that the intermediate minimum is not as distinct. For the captured floats, the dispersion differs significantly from the curves for all floats. During the first 13.5 hours, the dispersion of the isobaric floats ("drifters") increases from about 0.78 km to 1.17 km, and thereafter it decreases almost steadily to 0.45 km at the end of the integration. Thus, the drifters tend to converge, suggesting downwelling in the center of the spiral. In contrast, the dispersion of the isopycnal floats is rising from the initial value of 0.80 km to a final value of 1.51 km at the end, almost 4 times the values of the drifters. But the dispersion is not steadily increasing; instead, it attains a maximum of 1.35 km at 16 hours, thereafter it decreases to 0.87 km, and then it rises almost steadily, indicating that the floats depart from the center of the spiral.

The virtual drifters were released at 1-m depth along a zonal section across the northward extending filament, making them comparable to real drogued surface drifters. From the original ensemble of 134 drifters, a subensemble of 48 ones was captured by the orbital motion of the spiral within about 48 hours, while the remaining 86 ones moved into different directions. Correspondingly, the dispersion of the total ensemble increased during the first 30 hours, thereafter it decreased slightly and it increased again after 40 hours. Obviously, the decrease was caused by the clustering of the subensemble in the spiral (cf. Figs. 12c, d). This becomes more apparent, if the dispersion is computed only for the drifters in the cluster. Namely, during the first about 15 hours, the dispersion increases because the drifters are stretched along the umbilical of the spiral, but then it decreases rapidly, while the drifters accumulate in the spiral. The initial increase and the subsequent decrease closely resembles the dispersion of a real drifter cluster that was released by D'Asaro et al. (2018) in the Gulf of Mexico. But in contrast to the real cluster, the dispersion of the virtual ones decreased further and further, indicating that they are sucked in by the convergent flow and associated downwelling in the center of the cyclone. The latter, in concert with the convergence lines along the spiral arms, generates certainly the spiraliform streaks of floating materials like the cyanobacteria in Fig. 1.

The dispersion curve of the 50 isopycnal floats clustered by the flow in the spiral resembles the corresponding curve of the drifters, but the values are generally higher. The reason for this is that the isopycnal floats have to sink in response to surface convergence, and the sinking can only occur away from the center along the sloping isopycnals. The decreasing dispersion



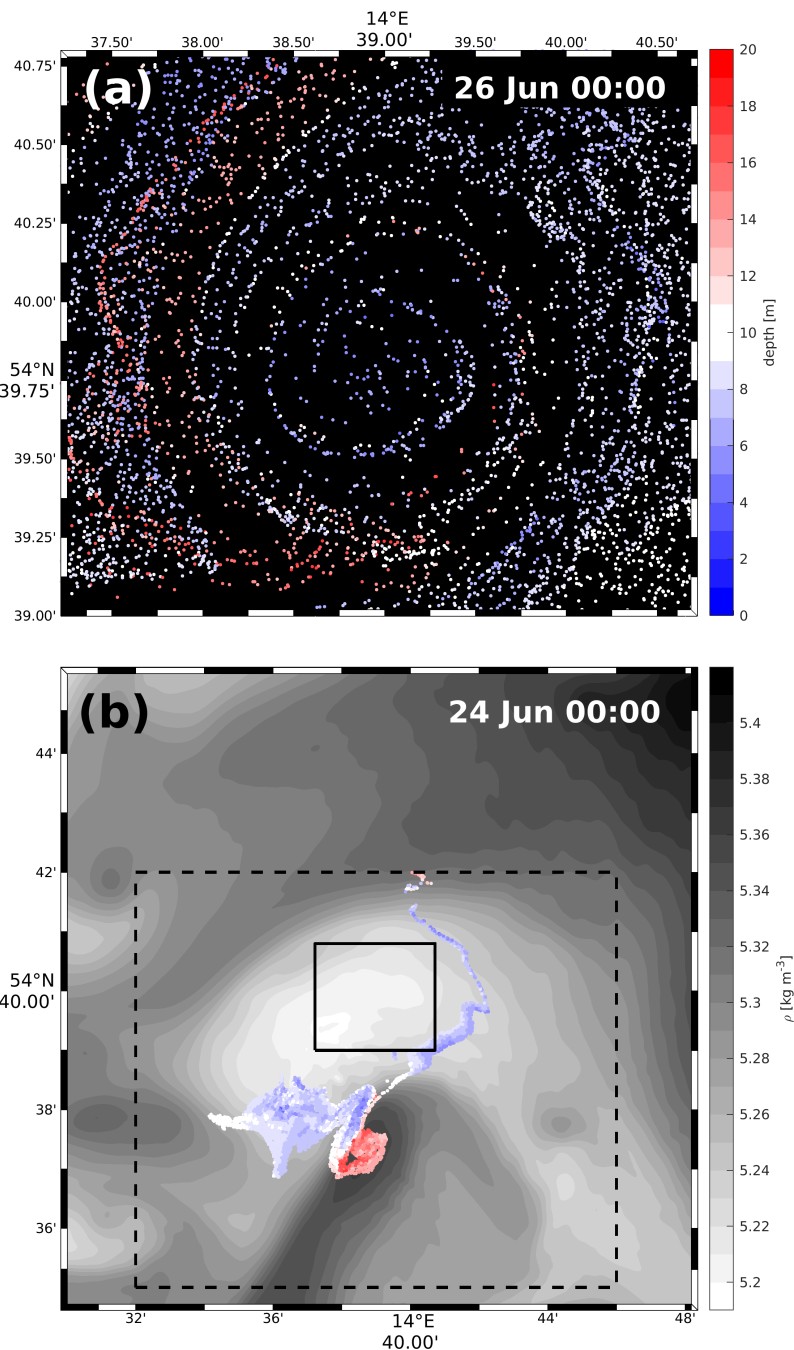

**Figure 21.** Experiment ISOPYC3: (a) Horizontal and color-coded vertical positions of floats on 26 June within the black frame shown in panel b, (b) 100,000 floats were randomly deployed at 10-m depth within the dashed rectangle on 24 June. The colored dots indicate the horizontal positions of the floats shown in panel a, using the same color code.





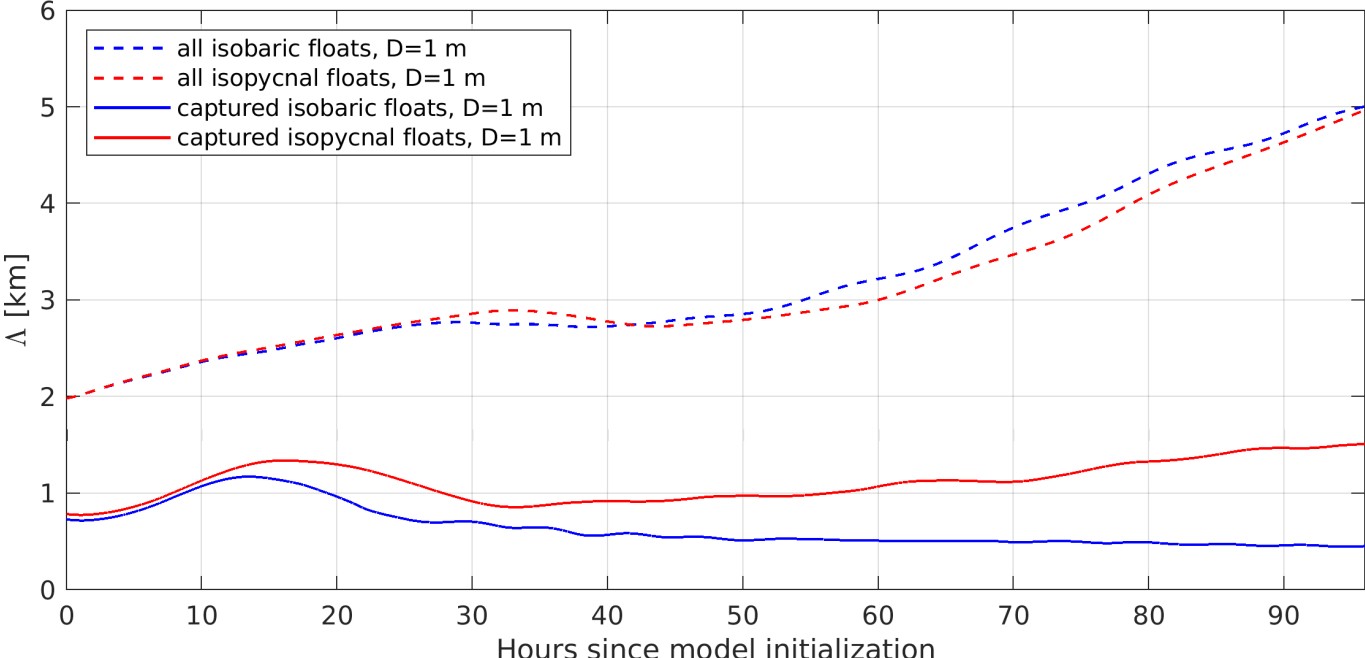

**Figure 22.** Dispersion of isobaric and isopycnal floats. Dashed lines refer to all floats, solid lines only to those ones that were captured by the cyclonic spiral. $D$ is the deployment depth.

between hours 20 and 33 is not in contradiction with that, because the majority of the Group I and Group II floats returns quite rapidly from their maximum depth to shallower levels, getting closer to the center. Moreover, instead of a further decrease, the dispersion of the shallow floats starts to increase again after 33 hours, which is most likely caused by Group I floats, sinking almost steadily during the first 3 days.

## 5    Discussion

### 5.1    Model setup

The evolution of a submesoscale cyclonic spiral was simulated by means of a triple-nested ROMS setup. This method is, of course, rather complex and expensive, but it allows to explore the kinematics of the spiral from the very beginning of a filament until it reaches a mature stage. Alternatively, one might have started from an idealized circular-shaped eddy like McGillicuddy (2015), but that would not reproduce the asymmetries of the coiling filament and the spiraliform vorticity. The latter, in turn, enforces non-zero vorticity advection and the associated VRWs, that finally shape the vertical motion pattern.

Atmospheric forcing, i.e. the surface fluxes of momentum, heat and freshwater, were turned off in R33 in accordance with the parent model R100. Realistic fluxes were only applied in the grandparent R500 in order to generate a mesoscale environment that is representative for early summer in the southern Baltic Sea. The turning-off is justified because mainly the momentum





flux impedes the generation of submesoscale instabilities (Renault et al., 2018; Kubryakov and Stanichny, 2015; Zatsepin et al., 2005) and blurs the corresponding structures as shown by Onken et al. (2020a) and Mahadevan et al. (2010). Different processes may lead to the generation of spirals, e.g. winding of filaments around existing vortices (Zhang and McGillicuddy, 2020), lateral straining of tracer fields (Meunier et al., 2019), or barotropic/baroclinic instability (de Marez et al., 2020). All

these processes are adiabatic, or at least, diabatic forcing is not required to get them going. Moreover, isopycnic potential vorticity is conserved under these condition, which facilitates the interpretation of the model results.

The Lagrangian floats in ROMS follow neutral surfaces (van Sebille et al., 2018), but our analyses of the individual properties of the floats are based on the assumption that the floats move along potential density surfaces. This is acceptable, because the differences between neutral surfaces and potential density surfaces are negligible, as long as the difference between the

reference pressure and the in situ pressure is small (You and McDougall, 1990). The latter applies for all floats. Nevertheless, as shown above (e.g. Figs. 16a, 20c), the density of several floats changes during the course of the integration, and concurrently the isopycnal potential vorticity. Thus, one is tempted to assume that ROMS does not conserve potential vorticity, but that would be a very strong statement and call the validity of the primitive equations or their finite difference equivalents into question. Instead, it is more likely that isopycnic potential vorticity is not conserved in a non-isopycnic coordinate system.

Other conceivable reasons could be the even though small diffusivity or that the float tracks are inaccurately computed, which makes them different from the paths of actual Lagrangian water particles.

## 5.2   Impact of downscaling

For the initialization of R33 on 24 June, the corresponding prognostic fields of R100 were mapped onto the R33 grid by linear interpolation. As the s-coordinates of the parent (R100) and the child (R33) were identical, the vertical interpolation did not

imply any approximations. In contrast, the mapping from the coarse-resolution horizontal parent grid on the higher-resolution grid of the child is a potential source of error, because the interpolation may generate local imbalances of the involved forces. Furthermore during the integration, boundary values were provided by R100 in 3-hour intervals along the open boundaries. As the baroclinic time step in R33 was 20 s, all prognostic variables along the open boundaries were updated by linear interpolation at each time step, which may create additional errors. In order to minimize such errors, it is recommended that the grid

refinement factor should not be much larger than 3 (McWilliams, 2016), which is the case in our setup. Nevertheless, the integration of the primitive equations may produce a solution in the child domain that is consistent with the initial and boundary conditions but deviates significantly from the solution of the parent. A tendency for R33 to develop its own solution, that is different from R100, is discernable in Fig. 3 already on 26 June, merely 2 days after the initialization. However, this does not necessarily imply that the solution is erroneous, particularly because there are no visible primary errors of downscaling

procedures, so-called rim currents (Mason et al., 2010) along the open boundaries. Moreover, the evolving spiral as the main object of our research is apparently not affected.

Despite of the potential shortcomings, the advantages of downscaling are striking, because the high-resolution child resolves features that are not visible in the solution of the parent. This is demonstrated by means of selected quantities in Fig. 23: the spiraliform shape of the surface density front in R33 is barely visible in R100 (cf. panels a, f); in the same way, R100 pretends





**Table 2.** Extreme values of various quantities in the top layer, except for $F$ that was evaluated at 2-m depth and $w$ at 5 m. The minima of positively defined quantities were left blank. The numbers in the last column are the ratios between the extremes in R33 and R100.

| Symbol | Meaning | Units | R100 Min | R100 Max | R33 Min | R33 Max | Ratio | |
|---|---|---|---|---|---|---|---|---|
| $\|\nabla\rho\|$ | Absolute horizontal density gradient | $10^{-4}$ kg m$^{-4}$ | | 1.8 | | 11.4 | | 6.3 |
| $F$ | Frontal tendency | $10^{-12}$ kg$^{-2}$ m$^{-8}$ s$^{-1}$ | -11.7 | 11.1 | -47.5 | 64.0 | 4.1 | 5.8 |
| $\|\boldsymbol{V}\|$ | Total speed | cm s$^{-1}$ | | 11.5 | | 12.9 | | 1.1 |
| $\|\boldsymbol{V}_{geo}\|$ | Geostrophic speed | cm s$^{-1}$ | | 11.5 | | 12.9 | | 1.1 |
| $\|\boldsymbol{V}_{ageo}\|$ | Ageostrophic speed | cm s$^{-1}$ | | 1.3 | | 2.7 | | 2.1 |
| $\zeta/f$ | Normalized relative vorticity | 1 | -0.7 | 3.6 | -2.2 | 12.3 | 3.1 | 3.4 |
| $\epsilon/f$ | Normalized horizontal strain rate | 1 | | 2.8 | | 18.2 | | 6.5 |
| $\delta/f$ | Normalized horizontal divergence | 1 | -0.5 | 0.6 | -3.0 | 1.8 | 6.0 | 3.0 |
| $w$ | Vertical velocity | m day$^{-1}$ | -12.9 | 12.4 | -80.8 | 60.4 | 6.3 | 4.9 |

a circular structure of the relative vorticity (panel c), that is a spiral as well (panel h). The dipolar character of the horizontal speed (panel g) is not reproduced by R100 (panel b), while R100 feigns a dipole of the frontal tendency (panel e), which is rather a multi-arm spiral with alternating frontogenetic and frontolytic sectors in R33 (panel j). Similarly, the tight sequential arrangement of up- and downwelling cells (panel i) is not at all resolved in panel d. Moreover, in Table 2, we have collated the

extreme values of various near-surface quantities and computed the ratio of the extremes between the child and the parent. A large ratio of 5.8 was obtained for the frontogenetic tendency, being $64.0 \times 10^{-12}$ kg$^{-2}$ m$^{-8}$ s$^{-1}$ in R33 and just $11.1 \times 10^{-12}$ kg$^{-2}$ m$^{-8}$ s$^{-1}$ in R100. Hence, the child is almost 6 times more frontogenetic than the parent, which is also reflected by the 6.3 times higher ratio of the absolute horizontal density gradient. Similarly, this applies for the frontolytic tendency, where the ratio is 4.1. Large ratios of 4.9 (upwelling) and 6.3 (downwelling) also predominate for the vertical velocity, the normalized

horizontal divergence with ratios of 6.0 and 3.0 for convergent and divergent flow, and the normalized strain rate with 6.5. In contrast, smaller ratios between 3 and 4 are found for the normalized relative vorticity, while for the ageostrophic speed the ratio is even less at 2.1. Finally, the maxima of the total and the geostrophic speeds reach about the same magnitude. Thus, in addition to the higher resolution of various features, the downscaling enhances particularly the amplification and attenuation of fronts and the intensification of the vertical velocity field.

**5.3   Properties of the spiral**

The properties of the spiral were already compared with previous studies above. As there were no numerical models known to the authors focusing explicitly on the properties of submesoscale eddies and least of all of cyclonic spirals, the model results were checked against just 2 observational studies, namely those of Ohlmann et al. (2017) and Marmorino et al. (2018). These resembled astonishingly well the modeled patterns of relative vorticity, strain rate, and the radial structure of the horizontal





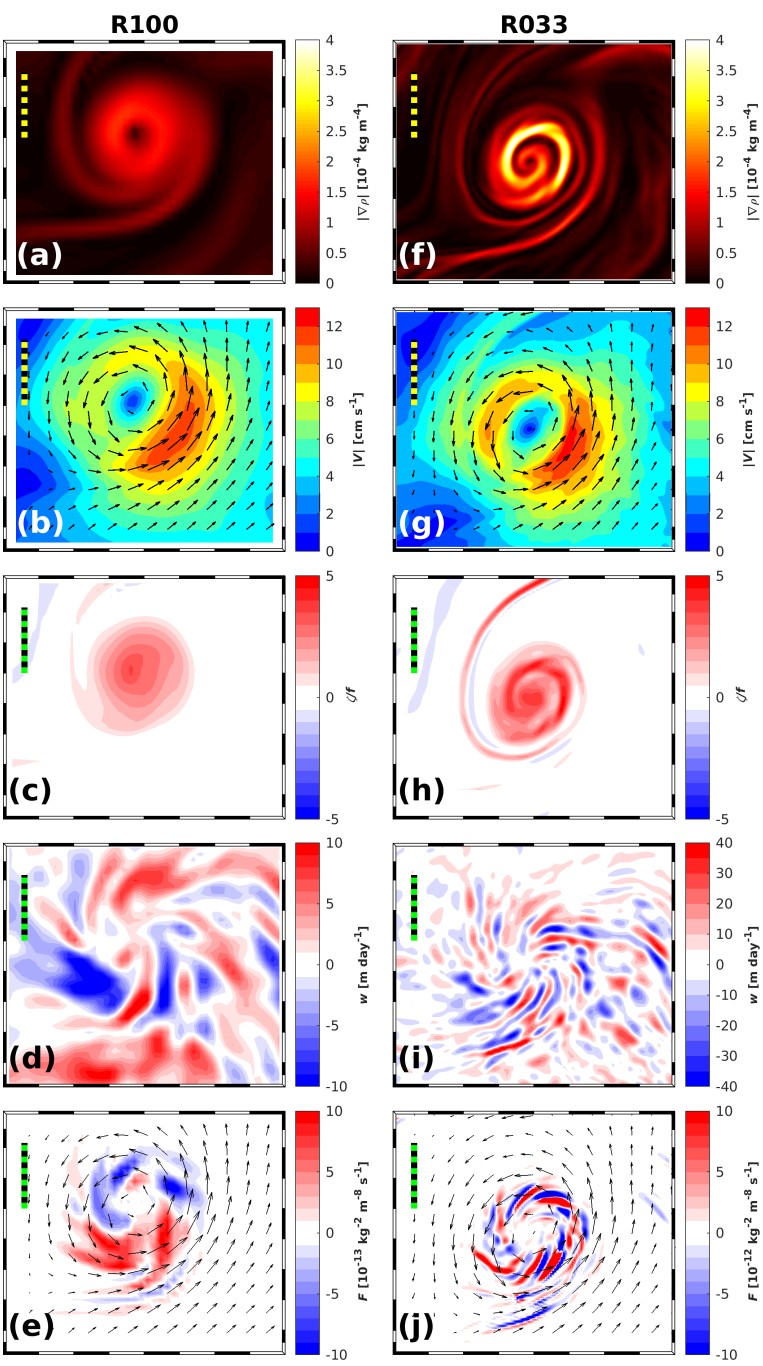

**Figure 23.** Selected near-surface properties of the spiral on 26 June in R100 (left column) and R33 (right). (a) and (f) $|\nabla\rho|$, (b) and (g) $|\boldsymbol{V}|$, (c) and (h) $\zeta/f$, (d) and (i) $w$, (e) and (j) $F$. Note the different color scales in (d) and (i) and the different scale factors of $F$ in (e) and (j). $|\nabla\rho|$, $|\boldsymbol{V}|$, and $\zeta/f$ were evaluated in the top layer, $w$ at 5 m and $F$ at 2-m depth.





divergence, but less similarity was obtained for the extremes of the divergence and the vertical velocity at the outer periphery of the spiral, that was estimated from the convergence of the outward radial flow. The remaining properties were left to be compared qualitatively with research findings related to mesoscale patterns and findings described in the meteorological literature. Agreement with other model studies was found with regard to the asymmetry of horizontal buoyancy gradients in the

filament and components of the tendency equation in the evolving spiral. Two or more maxima of the annular horizontal speed and multipolar patterns of the vertical velocity resembled both observed and theoretically derived structures. High-frequency azimuthal oscillations and the spiraliform arrangement of the vertical velocity due to VRWs exhibited striking similarities with results from other models, and the formation of secondary instabilities was validated by our own observations (Fig. 11). These agreements support the results of the present study.

## 5.4   Vertical motion

The sketch in Fig. 24 summarizes our findings with respect to the vertical circulation during the formation phase of the spiral. In the near-surface layers, downwelling and corresponding subduction of isopycnals is invoked by the overall convergent flow as suggested by the decreasing dispersion of the drifters captured by the spiral (Fig. 22). The mean vertical velocity was estimated from the vertical displacements of isopycnal floats during the first 2 days. It exhibits radially arranged bands with enhanced

downwelling, that are congruent with the spiral of elevated cyclonic vorticity; hence, the downwelling is apparently forced by material changes of the relative vorticity, which cause vortex stretching (Fig. 17) and associated convergent horizontal flow. According to Figs. 16b, and 20a, maximum downwelling of up to 18 m day$^{-1}$ prevails in the center of the spiral for the "shallow" floats that were deployed at 1-m depth; the "deep" floats deployed at 10-m depth descended at about half that speed (Fig. 21a). Downward velocities of the shallow floats up to 10 m day$^{-1}$ are also found in the bands with enhanced downwelling,

while the downwelling between the bands is rather modest, reaching not more than about 5 m day$^{-1}$. In contrast and in order to compensate for the subducted amount of water, the peripheral deep floats rise at a about the same speed, opposing the sinking shallow floats from above. However, it is not clear whether the shallow and the deep floats really converge in the same isopycnal layer or whether they glide on top of each other as suggested by Fig. 7m. One should also bear in mind that the circulation scheme in Fig. 24 is based on mean vertical displacements. In reality, the instantaneous vertical motion pattern in

the spiral is organized more slab-like with neighboring up- and downwelling cells as shown in Fig. 7h. These slabs are created by VRWs with a revolution period around the eddy center (estimated to 40 hours, see above Section 3.6) that is much larger than the orbital period of about 10 hours of a water particle at 500-m radius, assuming an azimuthal speed of 10 cm s$^{-1}$. Hence, each Lagrangian particle experiences high-frequency vertical oscillations along its trajectory as shown in Fig. 17. While both the shallow and the deep peripheral floats continue to orbit the vortex during their vertical displacement, those starting in

the density maximum of the filament feature the same capability to escape immediately from the cyclonic circulation already during the first day. According to the trajectories of the runaways, they slip rapidly down along isopycnals and thereafter tunnel under the swirling flow in the upper layers.

In a nutshell, downwelling prevails near the surface and in the center of the spiral at all depth levels. It is driven by convergent near-surface flow and associated subduction of isopycnals. Compensating weak upwelling is found above 10-m depth at



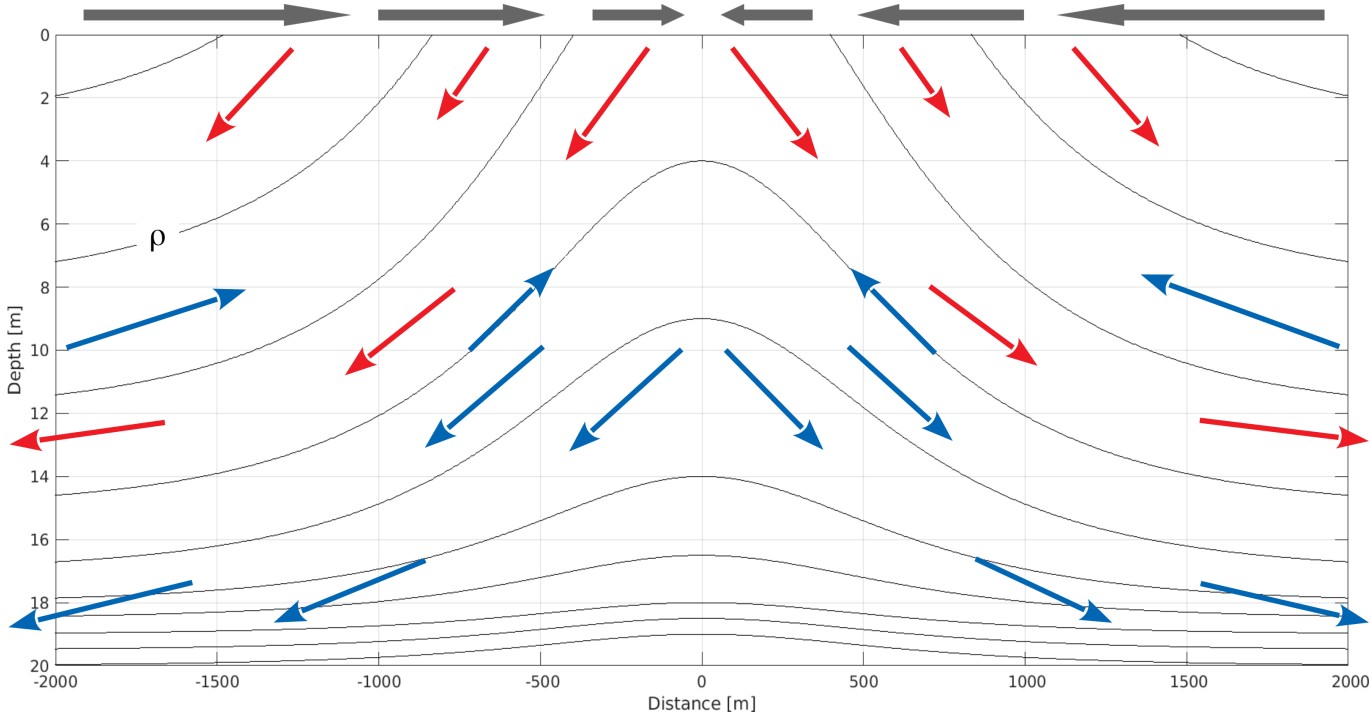

**Figure 24.** Mean displacements of Lagrangian particles during the spin-up phase of the spiral. The displacements of particles released initially at 1-m depth and 10-m depth are tagged by red and blue arrows, respectively. Gray vectors indicate the convergent near-surface flow.

the periphery of the spiral. This scheme is consistent and plausible but it contradicts the findings of other authors. For instance, Buongiorno Nardelli (2013) in his numerical study of a mesoscale cyclone found "...an average upwelling in the eddy core, downwelling at the kinetic energy maximum and upwelling again at the eddy periphery", and from the observations of a sub-mesoscale eddy by Marmorino et al. (2018), a radial outward flow at the surface and convergence and associated downwelling

was diagnosed along the eddy perimeter. In contrast, convergent near-surface flow in cyclonic eddies was found in the numerical experiments of Zhurbas et al. (2019) with synthetic floating Lagrangian particles embedded offline in a high-resolution circulation model of the southeastern Baltic Sea. While in anticyclones, the particles moved away from the center, particles in the cyclone exhibited the tendency to shift towards the center. Real surface drifters were tracked in the observational study of D'Asaro et al. (2018), who examined a cyclonic eddy a few kilometers in diameter. The drifters were spiraling inward the

eddy and computations from the eddy tracks yielded definitely convergent flow in the eddy core. Also in the sea ice-ocean model of Manucharyan and Thompson (2017), the surface convergence was found to be the primarily responsible process for the accumulation of ice in submesoscale cyclonic eddies in the marginal ice zone. The associated intense downwelling in the center and the weaker and broader upwelling in the surroundings agree perfectly with our model results. The physical reasons for the conflicting vertical circulation schemes in cyclonic eddies are discussed by Flierl and McGillicuddy (2002) (their Fig-

ures 4.8 and 4.21) and Bakun (2006) who distinguished between forced ("spinning up") and free ("spinning down") eddies





in a two-layer model: in case of a forced cyclone, the frictional torque is positive. This effectuates an intensification of the cyclonic vorticity which in turn leads to upwelling in the eddy core and associated divergent flow at the sea surface. On the other hand, in a free cyclone, that is not trapped by topography or coasts, the frictional torque is negative, and the diminishing cyclonic vorticity forces downwelling in the core with convergent surface flow and an outward directed flow in the lower layer.

Analogously, the circulation is inverse in anticyclones as confirmed by Bashmachnikov et al. (2019).

In R33, the evolution of the spiral is tracked over 4 days, starting from the very early stage of a dense filament on 24 June. During the first day, the filament is rolled into a vortex, and the roll-up is completed in the morning of 25 June when the density anomaly of the spiral separates from the filament. Subsequently, the mature spiral is tracked until 28 June. Based on these heuristic arguments, one may denote therefore the first about 30 hours as the spin-up phase of the spiral. However, the

modeled spiral during this period of time is not a "spinning-up" cyclone with upwelling in the center as defined by Bakun (2006). That type of cyclone requires external frictional forcing which is absent in the modeled cyclone. Instead, patches of up- and downwelling alternate due to the convergent spiral arms of $w$ (Fig. 7), and the mean direction of the vertical velocity is downward. Hence from the beginning, the modeled spiral is a free cyclone, that exhibits the characteristics of a "spinning-down" cyclone. This is also consistent with the subduction of isopynals (cf. Figs. 8k–o and 19), that starts already in the early

morning of 24 June, i.e. the isopycnals where never lifted up ("eddy pumping") as claimed by Flierl and McGillicuddy (2002) and McGillicuddy (2016).

In the light of the findings above, one may reinterpret the only existing observational study of a submesoscale spiral, which appears to be in conflict with our results. That spiral was observed by Marmorino et al. (2018) in the Southern California Counter Current (SCCC) at the northern tip of Santa Catalina Island (see the "study area" in the inset of their Figure 1).

Assuming that both our results and the observations are not faulty, we are left to interpret the different circulation patterns in terms of the schemes of Flierl and McGillicuddy (2002) and Bakun (2006). Accordingly, the modeled spiral is a free cyclone, while the secondary circulation of the observed eddy exhibits typical characteristics of a forced cyclone. As shown by the model experiments of Dong and McWilliams (2007), the forcing is due to lateral and bottom stresses, forming sheets of positive vorticity and associated current-wake instability (Marmorino et al., 2010) when the SCCC flows along the northeast

coast of Santa Catalina Island. A part of the sheet leaves from the northern tip of the island and transforms into a circular cyclonic spiral and continues to propagate downstream with the SCCC.

## 5.5 Implications

Our model results characterize the ageostrophic secondary circulation in an evolving submesoscale cyclonic eddy. In analogy to dense filaments (McWilliams et al., 2009; McWilliams, 2017), the secondary circulation is an overturning cell with intense

downwelling in the eddy center and weak upwelling at the periphery. Besides the ability to accumulate ice (Manucharyan and Thompson, 2017), the convergent near-surface flow may concentrate any other buoyant flotsam in submesoscale cyclones, such as plastic debris (van Sebille et al., 2020; Barboza et al., 2019; Turner et al., 2019), oil (D'Asaro et al., 2018), macroalgae (Zhong et al., 2012), buoyant plankton (Hernández-Hernández et al., 2020), or cyanobacteria (Marmorino and Chen, 2019). However, the convergence is neither uniformly spread across the cyclone nor concentrated in a single point in the center;



instead, it is organized in multi-arm spirals with negative helicity comprising alternating bands of convergent and divergent flow. As downwelling is intimately tied to convergent flow, it affects only particles with negative or zero buoyancy. According to Gemmel et al. (2016) and Lännergren (1979), the majority of phytoplankton species are negatively buoyant, although there are a few exemptions (Woods and Villareal, 2008). Hence during the plankton bloom, when the nutrients in the near-surface

layers are depleted, phytoplankton cells at the periphery of the spiral may sink by a few meters within a weakly stratified water column and find more favorable conditions, particularly because of the upward transport of nutrients on the same isopycnals (Fig. 24). Potentially, this mechanism contributes to the formation of chlorophyll rings in mesoscale eddies as observed by Xu et al. (2019). In contrast, cells in the center of the spiral reach depth levels between 15 m and 20 m, i.e. within the seasonal thermocline. This eddy-driven subduction may support the bloom of cyanobacteria at deeper levels in July and August (Hajdu

et al., 2007) and beyond that it reinforces the export of particulate organic carbon from the mixed layer (Omand et al., 2015; von Appen et al., 2018).

## 5.6 Transferability

Details of the modeled spiral were partly compared with the corresponding properties in mesoscale eddies. This was necessary because of the lack of information with respect to the appropriate submesoscale features, both from observations and models.

As frequently sufficient agreement was found, this raises the question whether and to what extent our findings can be conversely transferred to structures and motions in mesoscale eddies. To begin with mesoscale cyclonic spirals, it is unclear how often they occur. Namely, by means of SAR (synthetic aperture radar) images, Dokken and Wahl (1996), DiGiacomo and Holt (2001), Karimova and Gade (2016), and Karimova (2012) detected thousands of cyclonic spirals in various waters, but the diameters of the vast majority (partly more then 99%) was less than 20 km, suggesting that most of the spirals are submesoscale features.

However, that does not mean that mesoscale spirals do not exist; potentially, they are more difficult to discover with SAR because the required changes of surface roughness are too weak on larger scales. In contrast, by combining ocean-color, satellite altimetry and surface drifter data, Zhang and Qiu (2020) demonstrated that spiral bands of chlorophyll enhancement emerge globally both in cyclonic and anticyclonic mesoscale eddies, and Zhang and McGillicuddy (2020) verified the existence of spiraliform streamers in Gulf Stream anticyclonic rings by means of satellite-measured sea surface temperature. Hence, both

submesoscale and mesoscale spirals appear to be common features in the ocean. However, submesoscale spirals are cyclonic, nearly without exception, and they develop from a filament which is rolled in a spiraliform vortex. By contrast, mesoscale spirals are both cyclonic and anticyclonic, and they are formed in rings that have been detached from unstable jets. Although the forms of appearance of all spirals are very similar or even identical, this does not necessarily mean that the dynamics are the same, because the balance of forces is different for submesoscale and mesoscale motions.

## 30 6 Conclusions

The evolution of a small submesoscale cyclonic spiral is simulated by means of a high-resolution circulation model. The generation of the unforced spiral starts from a dense filament that is rolled into a vortex and becomes detached from the

filament. The first about 36 hours of that process are referred to as the formation or spin-up, and the time after as the decay or spin-down of the spiral. Here, the meanings of "formation" and "spin-up" differ from the definitions of Flierl and McGillicuddy (2002) and Bakun (2006), who used the identical wordings for forced eddies.

The spin-up is accomplished by a pre-existing mesoscale circulation pattern, that transforms the straight filament into a
spiral. During this phase, all-time extreme values are attained by various quantities. While some become organized in single-arm spirals, multi-arm spirals with alternating signs are characteristic patterns of divergence, frontal tendency, ageostrophic, and vertical velocity. The multi-arm spirals are forced by vortex Rossby waves, that are excited by advection of vorticity. The ageostrophic together with the vertical velocity effectuate the overall decrease of baroclinicity and contemporaneous subduction of isopycnals and mixed-layer shoaling, indicating baroclinic instability as the main driver for the formation of the spiral.

The spin-down starts when the cyclone separates from the filament. At the same time, the horizontal speed develops a dipole-like pattern and isotachs generate closed contours around the center of the vortex. The amplitude of most quantities decreases significantly, except for the instantaneous vertical velocity, that attains even more pronounced extremes than during spin-up. Secondary instabilities in the wake of the spiral are potential gateways for the cascade of energy towards the microscale.

The mean ageostrophic secondary circulation comprises a circular symmetric overturning cell with intense downwelling in
the center and weak upwelling at the periphery. The associated convergent flow at the surface may concentrate buoyant flotsam in the spiral. The downwelling in the center reinforces the export of particulate organic carbon from the mixed layer into the main thermocline. The upward isopycnal transport of nutrients at the periphery supports the growth of phytoplankton in the euphotic zone.

*Code and data availability.* The model code and the output of the ROMS runs presented in this article are available from the first author on
request.

*Author contributions.* Reiner Onken has set up and analyzed the model studies and has written the manuscript. Burkard Baschek has interpreted the results in the context of submesoscale eddy dynamics and the connection to the "Expedition Clockwork Ocean" that motivated this study. He also edited the manuscript and provided Figures 11 and 24.

*Competing interests.* The authors declare that they have no conflicts of interest.

*Acknowledgements.* The present study was funded by Helmholtz-Zentrum Geesthacht/Hereon through the program of the Helmholtz Association PACES II (Polar regions and coasts in the changing earth system). The authors are grateful to Daniel Behr and Hajo Krasemann, who accomplished the georeferencing of Figure 1.



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
