# Peer review of "Properties and evolution of a submesoscale cyclonic spiral"

_Ocean Science, 2021_

## Author Comment (AC1)

Comments of Reviewer1 are blue.
Authors' reply are black.
* * *
A review of "Properties and evolution of a submesoscale cyclonic spiral" by R. Onken and B. Baschek submitted to Ocean Science.

General comments:

The manuscript describes results from a triple-nested configuration of a submesoscale cyclonic eddy in the Baltic Sea. The simulation is set up to represent observed situation from 2016 and aims at elucidating evolution of the cyclonic feature.
The paper addresses a couple of very relevant questions within ocean science, namely model representation of (sub-)mesoscale eddies in the Baltic Sea, and the evolution and dynamics of these eddies.
This is not quite correct. Only one cyclonic spiral is investigated in detail. See ms P4L4–5.
The triple-nested Regional Ocean Modelling System configuration is novel and ambitious. However, I find significant shortcomings regarding the experiment design, analysis and presentation of the results, as well as text organization and writing. It is difficult to understand what conclusions of the study are. I recommend rejection and optional resubmission after a major reworking of all the aspects of the study. Below I am highlighting the major issues, noting that it is not practical to comment on all the details at this stage.
Manuscript structure and writing:
The organization of the text and writing make reading and review of the content difficult. It is hard to disentangle a thread of the narrative as well as the aims, objectives and conclusions of the study. The parts pertaining to Methods are interspersed with and within other sections of the manuscript. The Introduction starts with a vague description of the background, emphasizing lack of understanding of the evolution and dynamics of the submesoscale eddies and in particular cyclonic spiral-like eddies (pages 1-2) - however, without a definition of the "submesoscale" and "mesoscale" notions in the Baltic Sea region(!) - these two terms are usually defined in terms of the first Rossby radius of deformation (that depends on the local stratification and latitude-dependent Coriolis frequency) - no comment on this or any alternative definition is offered.
Concerning the definition of "mesoscale" and "submesoscale", the reader is referred to the companion paper Onken et al. (2020), see P4L3. Moreover, "mesoscale" and "submesoscale" are not defined by means of the Rossby radius but by the Rossby number and the Richardson number! For details, see as well Onken et al. (2020) and McWilliams (2016).
Neither is it clear what is actually meant by "physical properties", "evolution" or "dynamics" in the text (neither of these notions being sufficiently quantified by the analysis, see below).
Section is dealing exclusively with physical properties and the evolution of the spiral (P5-P18=13 pages!). The dynamics (=interplay of forces) is referred to in Sections 3.1, 3.6. And the concept of the conservation of isopycnal potential vorticity in Section 4 (P18-P21) is pure dynamics!
On page 4 of the Introduction we find some details about the model configuration (which pertain to the Methods section 2) together with the information that the model "creates the mesoscale background for June 2016" ("mesoscale" being undefined) with times reported in UTC, but it is not clear whether the study aims to realistically reproduce the observed situation that day or is meant as

a more general modelling study? I could not find answer to this question after reading the entire manuscript.

- P2L34-P4L5 together with P1L16 clearly describes the objective of the study.
- It is unclear to us what you mean by "observed situation that day" and "more general modelling study".
- Section 2 is not about methods. Instead, it provides a short description of the ROMS model, the formation of the spiral in R100, and the downscaling in R33.

Some more details missing int the Methods section are further found in the Result section 4 (e.g., Table 1 showing the setup of the experiments).

- There is no "Result section 4". Section 4 is about drifters and floats
- Table 1 refers only to the setup of the experiments with drifters and floats.

A summary of the numerical Results (Table 2) is found in sect. 5 (Discussion) while it fits better in the Results section.

- Again, there is no Results section.
- Table 2 belongs to the Discussion 5.2

The final sections Implications (5.5) and Transferability (5.6) miss the targets. It is incorrect to compare the results from the Baltic Sea to other regions where submesoscale eddies have been reported without a careful discussion of the different dynamical regimes (and different scales for submesoscales).

- Why is Section 5.5 missing the target? The implications of our findings related to the secondary ageostrophic circulation are (i) the ability to accumulate ice, (ii) to concentrate buoyant flotsam by convergence, (iii) the sinking of phytoplankton, and (iv) the upward transport of nutrients.
- The target of Section 5.6 is clearly defined on P44L15-16.
- Once again, the submesoscale regime is not defined by the scale but by the Rossby number and the Richardson number.

A discussion of the relevance of the study as well as the implications of the existence of the cyclonic spirals for the Baltic Sea region are entirely missing(!).

This was not at all the objective of the study!

Some of the sentences and expressions are very intricate or unspecified ("submesoscale waveband", "the perceptions are rather fragmentary", "downscaling appears to be perfect", "the nesting procedure does apparently a good job"). An adverb "hence" (as a consequence; for this reason) is used multiple times while the logical consequence is not always clearly emerging from the preceeding statement.

- "Submesoscale waveband" is described in many articles (e.g. see the pertinent papers of McWilliams and coauthors)
- "perceptions are fragmentary" (on P2L15) means that the 3 citations provided only a limited description of the physical properties of a spiral.
- "downscaling appears to be perfect" …. because no primary errors are visible, e.g. rim currents (see P38L30).
- "the nesting procedure … good job": same as above
- "hence": we will check

Methods and analysis:

Several important details about the model configuration are missing which undermines confidence in the results.

Which important details do you mean?

The manuscript is very long and contains long descriptive passages or visualizations of the model simulations which bring in very little quantitative and qualitative information (e.g., a very detailed description of the Lagrangian simulations in sect. 4).

Section 4 is the core of the paper! It demonstrates the impact of the kinematical and dynamical (=physical – see above!) properties on the behavior of Lagrangian drifters and floats. See P4L7-8.

The reported values of e.g., velocity or frontal tendencies are hard to interpret without a reference to expected values for this (regional!) regime and information what was the sampling (averaging) frequency from the model (are snapshots the daily or hourly averages?).

All figures are snapshots. We might have missed to say that – agreed!

The 23 (!) figures are merely visualizations of the model simulations while the analysis and statistical quantification of the results is nearly entirely missing.

- Why "merely"?
- A statistical quantification was never intended. However, some statistics is provided by the experiments with 100,000 particles.

Some of the inferred interpretation (i.e., about the Vortex Rossby Waves or types of instabilities) is not (or at least not sufficiently) supported by the results. The references are cited somewhat selectively and not always aptly in various places.

- The vorticity advection in Fig. 9d proves the existence of VRWs
- What "types of instabilities" do you mean?
- What do you mean by "selectively"? Not adequately or arbitrarily?

Below I include a non-exhaustive list of questions and issues that need to be clarified in addition to the points raised above. I am not including new references to the existing list since a lot of information and suggestions for analysis and diagnostics are already found in the papers cited by the manuscript, but they have not been used.

What suggestions for analysis and diagnostics do you mean? Please specify!

Methods:

- Already mentioned, missing definition of the mesoscale and submesoscale in the Baltic Sea and physical background of the study (local stratification, shallow basin, turbulent regimes). As the definition of (sub-)mesoscale is not universal, the direct comparison with studies in other regions is not appropriate unless a care in interpretation of the differences in dynamical regimes is taken.

See our replies above.

- Are spiral cyclonic eddies routinely observed in the Baltic Sea, how often, in what parts of the sea? How universal or specific is the study?

An answer to this question is beyond the goals of this study.

- Is the vertical resolution of 10 layers sufficient to quantify submesoscale dynamics in the region?

It is sufficient, because the water depth in R33 is around 40 m; see Fig. 2.

Is the vertical resolution uniform, is the haloclince/pycnocline sufficiently resolved?

S-coordinates are in general not uniform. For the parameters of the vertical grid see Onken et al. (2020a). Higher resolution is specified near the surface and near the bottom.

- Is the resolution of 33m justifying the hydrostratic approximation of the model?

It is not yet clear at what scales non-hydrostatic effects begin to become important. From a personal communication with Dewar, Gula, Molemaker (2018), 10 m appears to be an appropriate scale. Potentially, a repetition of this study with CROCO might give an answer, but CROCO is still in an experimental stage.

- Is the analysis performed on the snapshots or model output averages (daily or hourly averages)? This will certainly impact the (extreme) numerical values reported in e.g., sect 3.1-3.2.

All figures are snapshots. We have missed to say that – agreed!

- In sect. 4 we find a piece of information "As atmospheric forcing is turned off, the dynamics of R33 can be considered as adiabatic and nearly frictionless because no explicit vertical mixing is specified, and the biharmonic diffusivity coefficient is extremely small".  First, this information pertains to Methods section,

A separate section describing all applied methods does not exist and it was never intended to create such a section. All methods are introduced at the instant when they are needed.

second, it should also be further elucidated. As the atmospheric forcing is not turned off in the parent simulations (if I understand correctly), is R33 indeed adiabatic (how the nesting procedure carries)?

Atmospheric forcing is only active in R500 (the grandparent).

What is the mechanism for the MLD changes reported in sect. 3.5 if the atmospheric forcing is entirely off? Is this realistic?

Under adiabatic conditions, MLD changes are only forced by vortex squeezing/compression and by restratification due to mixed-layer (baroclinic) instability.

- Results about the frontal tendencies (sect. 3.1.) hard to interpret. Please see Capet etal 2008 for more details. The fact that some other studies in other regions like coast off California showed that the major contribution to the frontogenetic tendency comes from the straining deformation of the horizontal velocity, while the vertical straining is the main driver of frontolytic processes, does not mean that this is also the case for the presented study in the Baltic Sea. Please verify the statement by appropriate diagnostics (Capet). What is the underlying theoretical explanation and meaning of this result?

It is not the intent of the authors to deepen the diagnostics of the tendency equation at this place. This has been discussed at length in Onken et al. (2020).

- Sect. 3.2 and Fig. 5. Reporting the extreme values of geostrophic vs ageostrophic velocity with numerical values is superfluous if the local Rossby number is also used (also values of the velocities or the "existence of multiple maxima" compared to other studies is not meaningful). Section 3.2. c be entirely removed.

One of the major goals of this article is to describe the physical properties of the spiral. The horizontal velocity and the relative contributions of the geostrophic and ageostrophic parts are key properties.

- Sect. 3.3. Please explain what is the rationale of using the vertical component of the relative vorticity only in the presence of the frontogenesis processes.

We do not understand the question. The formation of the spiral is controlled by vorticity and strain, and their relative contributions are expressed by the Okubo-Weiss parameter.

- Sect. 3.3. Please justify the statement: "While in adiabatic flow, $\zeta > -1$ should hold, $\zeta < -1$ is probably a consequence of the diabatic contribution from the biharmonic diffusivity, caused by the extreme horizontal density gradient" This can be verified by comparing the contribution of the terms in the vorticity evolution equation. Is the "diabatic contribution from the biharmonic diffusivity" an artifact of the numerics? I do not quite understand this statement as the triple nest simulation was reported in Sect. 2 to have no explicit mixing?

"No explicit mixing" (P5L7) refers only to the mixing of momentum, i.e. the eddy viscosity in R33. In contrast, the biharmonic mixing refers to the diffusion of tracers.

What is the (dynamical, theoretical) interpretation of the evolution showed in Fig. 6? The statement: "indicating a bimodal structure with extremely strong control of both vorticity and even more strain" is vague and brings in no quantitative information.

This is not correct! The authors say that the "… Okubo-Weiss-parameter … exhibits the largest spread on 25 June with 106 < eta_prime < 258" (P12L1), and this is definitely a quantitative information.

One could present PDFs of vorticity/strain

Why PDFs? The relative contributions of vorticity and strain are exactly expressed by Okubo-Weiss.

or vertical profiles of Kurtoses/Skewnesses instead and interpret the information.

Our analyses of the physical properties of the spiral (except for the vertical motion and the stratification) are restricted intentionally to the top layer.

What is the meaning of comparing these values to the study in the Gulf Stream (Gula) - rings vs spirals and different dynamical regimes?

We just say that the structures of the Okubo-Weiss parameter in Gulf Stream rings and in our spiral exhibit similarities. The dynamical regimes are not different: both are submesoscale, as the local Rossby number (= f-scaled relative vorticity) is O(1).

- Sect. 3.4. starts with: "In ROMS, the integral is actually computed as a sum from the bottom upwards and also as a sum from the top downwards, resulting in a linear combination of the two, weighted so that the surface down value is used near the surface while the other is used near the bottom (Hedström, 2018). Thus, the near-surface vertical velocity largely reflects the divergence pattern in the surface layer..." This information pertains to the Methods section.

Once again – a special section on methods does not exist, and it was never the intent of the authors to create such a section. In this article, the applied methods are introduced when they are needed; this is our style of writing and it should be accepted.

I do not quite understand how the sentence after "Thus" results from the preceeding statement, and what the statement after "Thus" implies - what else should the near-surface vertical velocity reflect?

In a hydrostatic model, the vertical velocity w is a diagnostic quantity that is computed by means of the vertical integration of the horizontal divergence. The results of this computations depend on the bounds of the integral, i.e. when integrating from the bottom to the surface (assuming w=0 at the bottom) is different from the result if we start the integration at the surface (w=0 at the surface).

Also I do not quite understand why (Hedström, 2018; technical manual for OCS BOEM study) is cited to support a statement about the vertical velocity computation in ROMS. If this paragraph is meant to cast a confidence on the model vertical velocity output, it has had exactly an opposite effect on me...

Presently, Hedström (2018) is the most complete description of the ROMS model. It is considered as **the** ROMS manual.

The text that follows, and Figure 7 ate too descriptive. Please quantify the "progressive decorrelation" or quantify by time scales w by e.g., spectral analysis and interpret. The spatial pattern of w is not quantified either. What processes create these patterns? Please see the cited references at the end of sect,.3.4. for tips on relevant diagnostics.

- "too descriptive": before interpreting the vertical motion pattern, it has to be described
- "progressive decorrelation": we think that the description on P12L28-32 is sufficient.
- "spatial pattern of w is not quantified": how should a spatial pattern be quantified?
- "What processes create these patterns"? An explanation is provided in Section 3.6

- Sect. 3.5 + Fig.8. Vertical stratification, focus on MLD. Hard to interpret the absolute numbers reported, can be skipped or information-compressed using statistical diagnostics and/or vertical profiles.

This description is absolutely necessary in order to understand how the MLD is modified during the spin-up of the spiral.

What processes generate the variations of the MLD? I don't understand the immediate relevance of the statement: "The subduction of the 5.3 kg m −3 isopycnal in the center, the contemporaneous rising at the periphery of the spiral, and the leveling of the MLD are clear indicators for restratification by mixed-layer instability (Boccaletti et al., 2007; Fox-Kemper..." This should be verified for the particular regime relevant to the regional study presented and particular simulation with appropriate analysis (see cited references).

The existence of mixed-layer instability is verified by subduction and restratification. The reviewer might read the relevant literature.

- Sect. 3.6. Vortex Rossby Waves. Again, the relevance and transferability of the other cited studies in the ocean and atmosphere that interpret the spiral patterns of various quantities as VRWs is not immediate! Note that the interaction of the internal wave field with vortex field can also generate spiral features - the distinction between IWs and VRWs can be supported by e.g., estimation of the VRWs frequency from the dispersion relation and spectral analysis of w, in addition to analysis of the sources of w (see the cited references).

VRWs are driven by vorticity advection and the latter is  the major driver for the spiraliform vertical velocity patterrn (Fig. 9d). It is not known to the authors that the interaction of IWs with a vortex can generate such structures.

- Sect. 3.7. The origin of the secondary instabilities should be supported by appropriate diagnostics (e.g., energetics).

This would go beyond the scope of the paper.

This section can be also skipped unless its significance (and model realism) is discussed. A comparison based on visuals with results reported from the Gulf Stream not meaningful unless a full discussion of the dynamical regimes is given.

The authors just wanted to demonstrate that secondary instabilities exist both in the mesoscale (Ro<0.1) and submesoscale (Ro~O(1)) subrange. An explanation might be added.

A quantitative (model-data) comparison with observations from "Expedition Clockwork Ocean" is entirely missing (there is just a blurry figure 11 included).

This was never intended. The purpose of Fig. 11 is only to show that eddies with length scales of O(100m) are real in the corresponding area at the corresponding time.

- Section 4+5 (Lagrangian analysis). A more detailed information about the Lagrangian simulation technique (time step, in-line, off-line, volume/mass conserving or material points) missing which confounds the interpretation of the "isobaric" vs "isopycnal" property.

The simulation of isobaric and isopynal floats is a well established method in various ocean models. No more detailed information is necessary. See van Sebille et al. (2018).

Text in pPages 21-35 too descriptive and carry very little quanlitative and quantitative information. This entire part could be removed and a decent analysis in terms of absolute and relative dispersion in 2 and 3 dimensions should be included to quantify the transport properties associated with the spiral eddy.

This was never the intent of the authors. That would be a completely different paper!

Unclear statement on p. 38 ("floats in ROMS follow neutral surfaces" - aren't these actually s-surfaces in ROMS?

S-coordinates are terrain-following coordinates while neutral surfaces are isentropic surfaces.

- Section 5. Should be rewritten after addressing issues pointed out above.

No comment

Presentation quality:

The 23 figures should be critically revised since they contain superfluous descriptive/visual information while statistical/analytical information is missing. Please keep to good practices on colormaps outlined e.g., here: https://www.nature.com/articles/d41586-021-02696-z.

No comment

---

## Author Comment (AC2)

**Reply to the comments of Reviewer2 dated 23 November 2021**

Comments of Reviewer2 are blue.
Authors' replies are black.
* * *
This paper describes the evolution of a submesoscale spiraling eddy as simulated by a multiple nesting configuration of a ROMS numerical model set up for the Baltic Sea. The paper provides exhaustive analyses of the simulation, proposing interesting dynamical interpretations of the vortex oscillations and specific investigations on their impact in terms of vertical transport (carried out through Lagrangian approaches). The paper is written clearly, but it is quite long and the organization of the sections not always optimal. As an example, the model set-up is both described in section 2 and discussed again in section 5, which leads to repetitions and makes the reading much less pleasant.

Section 2 describes the technical details of the model setup, the downscaling procedure, and a rough description of the formation of the spiral in R100 and R33. In contrast, subsection 5.1 justifies the model setup and subsection 5.2 evaluates the impact of downscaling.

In subsection 5.1 (P37L12-P38L6), we justify the turning-off of atmospheric forcing, both in R100 and R33. This leads to adiabatic conditions and allows to interpret the model results under the assumption that isopycnic potential vorticity is conserved. In the last paragraph (P38L7-16), we discuss the fact that the Lagrangian floats in ROMS follow neutral surfaces, while our analyses re based on the assumption that the floats move along potential density surfaces. By comparing subsection 5.1 with section 2, we do not see any repetitions.

In subsection 5.2, there are indeed repetitions.

Suggested action:

The repetitions in 5.2 will be removed. Moreover, subsections 5.1 and 5.2 may be merged in a subsection entitled "Impacts of the model setup and downscaling".

I really suggest to significantly shorten it focusing on the most relevant findings and eventually removing at least some of the many details (and sometimes repetitions),

Agreed.

Suggested action:

Manuscript will be shortened.

also eventually moving some of the text to the figure captions (e.g.:"[…]The images are centered at the density maximum of the spiral, their meridional width is 2 arc minutes (=2 nautical miles ≈ 3704 m),[…]").

Agreed.

Suggested action:

We will check what pieces of text may be moved to the figure captions.

In many cases, instead of providing punctual descriptions of each figure within the text, it would be much more effective to directly focus on the relevant information the figure is giving, which would otherwise be missed by the reader.

Agreed.

Suggested action:

We will try to make the descriptions of the figures more concise.

My only concern about the scientific findings/discussion is related to the way eddy perturbations are identified as VRW. While I do agree that this is an absolutely plausible mechanism, some more comments would help to clarify how VRW can be distinguished by other vortex wave processes (e.g. inertial-gravity waves) and in case complemented by additional analyses that can reinforce the interpretation (e.g. by looking at propagation speeds predicted by VRW dispersion relation).

We are not experts on VRWs. We just detected the eddy perturbations in our animations of the spiral and conjectured that they are VRWs, mainly because of the similarities with observed phenomena in hurricanes.

Suggested action:

We will try to compute the radial and azimuthal phase velocities using equations (15) and (16) in Montgomery and Kallenbach (1997) or equations (3) and (4) in Chen and Yau (2000). These will compared with the corresponding quantities of inertial-gravita waves.

I would thus suggest a revision that could be classified between minor and major, but surely believe that the paper should be accepted for publication.